



# Upwelling-induced trace gas dynamics in the Baltic Sea inferred from 8 years of autonomous measurements on a ship of opportunity

Erik Jacobs[1], Henry C. Bittig[1], Ulf Gräwe[1], Carolyn A. Graves[2], Michael Glockzin[1], Jens D. Müller[1], Bernd Schneider[1], and Gregor Rehder[1]

[1]Leibniz Institute for Baltic Sea Research Warnemünde (IOW), Seestraße 15, 18119 Rostock, Germany
[2]Centre for Environment, Fisheries and Aquaculture Science (Cefas), Pakefield Road, Lowestoft, Suffolk NR22 0HT, UK

**Correspondence:** Erik Jacobs (erik.jacobs@io-warnemuende.de)

**Abstract.**

Autonomous measurements aboard ships of opportunity (SOOP) provide in situ data sets with high spatial and temporal coverage. In this study, we use 8 years of carbon dioxide ($CO_2$) and methane ($CH_4$) observations from SOOP *Finnmaid* to study the influence of upwelling on trace gas dynamics in the Baltic Sea. Between spring and autumn, coastal upwelling transports

water masses enriched with $CO_2$ and $CH_4$ to the surface of the Baltic Sea. We study the seasonality, regional distribution, relaxation, and interannual variability of this process. We use reanalysed wind and modelled sea surface temperature (SST) data in a newly established statistical upwelling detection method to identify major upwelling areas and time periods. Strong upwelling events are most frequently detected around August after a long period of thermal stratification, i.e. limited exchange between surface and underlying waters. We found that these strong upwelling events with large SST excursions shape local

trace gas dynamics and often lead to near-linear relationships between increasing trace gas levels and decreasing temperature. Upwelling relaxation is mainly driven by mixing and modulated by air–sea gas exchange and possibly primary production. Subsequent warming through air–sea heat exchange has the potential to enhance trace gas saturation. In 2015, quasi-continuous upwelling over several months led to weak summer stratification, which directly impacted the observed trace gas and SST dynamics in several upwelling-prone areas. We introduce an extrapolation method based on trace gas – SST relationships that

allows us to estimate upwelling-induced trace gas fluxes in upwelling-affected regions. In general, the surface water reverses from $CO_2$ sink to source and $CH_4$ outgassing is intensified as a consequence of upwelling. We conclude that upwelling is an important and relevant process controlling trace gas dynamics in near-coastal environments in the Baltic Sea, and that SOOP data, especially when combined with other data sets, enable flux quantification and process studies on larger spatial and temporal scales.

# 1   Introduction

The highly resolved quantification of dissolved greenhouse gases like carbon dioxide ($CO_2$) and methane ($CH_4$) across large spatial and temporal scales is critical to derive accurate climate projections (Friedlingstein et al., 2019) and helps to understand processes involved in the biogeochemical cycling of both gases (Takahashi et al., 2009; Webb et al., 2016). Traditional research

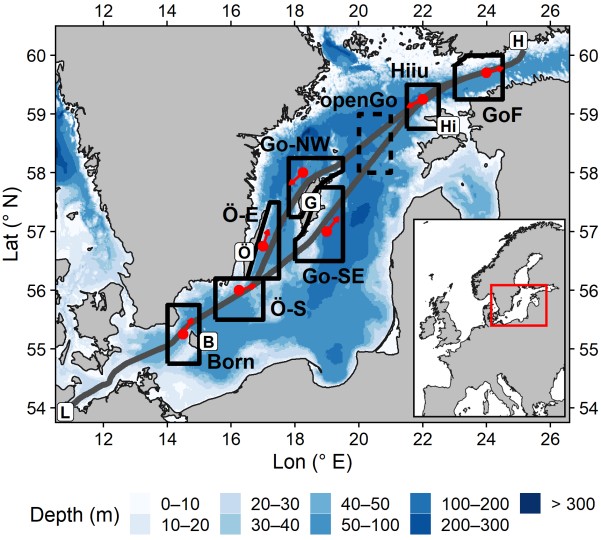

**Figure 1.** Map and bathymetry of the study area with typical routes of SOOP *Finnmaid* (grey) between Lübeck-Travemünde (L) and Helsinki (H). Boxes highlight seven regions (solid lines) in which we expect upwelling to occur and one in the open Gotland Sea (dashed lines) as reference (Table 1). Red dots mark the locations of wind data used for this study with red arrows indicating upwelling-favourable wind directions. We further marked the islands of Bornholm (B), Öland (Ö), Gotland (G), and Hiiumaa (Hi).

cruises usually involve discrete sampling in the water column and often continuous surface water measurements using an air–

water equilibrator coupled to nondispersal infrared spectroscopy for $CO_2$ (e.g. Körtzinger et al., 1996), gas chromatography for $CH_4$ (e.g. Bange et al., 1994), or cavity-enhanced absorption spectroscopy (CEAS, e.g. Gülzow et al., 2011; Du et al., 2014) – a relatively new technique with high sensitivity for both gases (Gagliardi and Loock, 2014). Whereas research cruises enable vertically resolved measurements in a certain region over several weeks, they do not provide wide temporal and sometimes spatial coverage, which is the main advantage of autonomous measurements aboard ships of opportunity (SOOP). SOOP

data sets enable studies with a broader perspective that – unlike remote sensing and modelling – are still based on in situ measurements. While SOOP-based $CO_2$ measurements are common owing to the available hardware (Pierrot and Steinhoff, 2019), $CH_4$ measurements are still scarce due to the comparatively recent development of CEAS and the elaborate setup aboard a SOOP (Gülzow et al., 2011; Nara et al., 2014). The ferry and SOOP *Finnmaid* hosts such an autonomous setup measuring surface concentrations of dissolved $CO_2$ and $CH_4$ since late 2009 using a CEAS sensor (Gülzow et al., 2011), resulting in a

unique long-term data set. The vessel transects the Baltic Sea between Lübeck-Travemünde (Germany) and Helsinki (Finland) about four times per week (Fig. 1). SOOP *Finnmaid* is part of the German contribution to the ICOS (Integrated Carbon Observation System) Research Infrastructure.

The Baltic Sea (Fig. 1) is a brackish, semi-enclosed sea in northern Europe characterised by riverine inflow of freshwater from a larger drainage basin and inflows of seawater from the North Sea, which, together with its basin structure, results in

a permanent vertical salinity and density stratification (Feistel et al., 2010). A shallow surface thermocline is present during


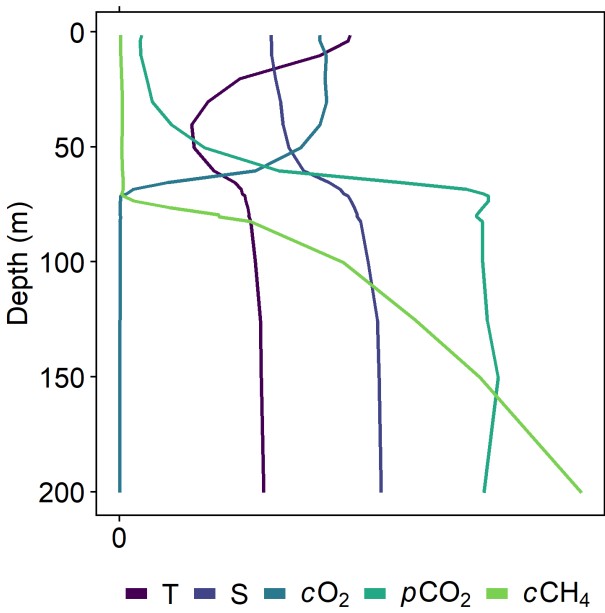

**Figure 2.** Typical vertical profiles of parameters relevant for this study in the central Baltic Sea in summer on arbitrary scale: temperature, salinity, oxygen concentration, $CO_2$ partial pressure, and $CH_4$ concentration.

summer (Fig. 2). The typical seasonality of surface $CO_2$ partial pressure ($pCO_2$) in the Baltic Sea features two minima in spring and summer: The spring bloom starts in March/April with the formation of the surface thermocline and peaks in mid-May (Schneider and Müller, 2018). Pulses of mid-summer bloom events are driven by nitrogen fixation (Schneider et al., 2014a). $pCO_2$ slightly increases in between due to warming, air–sea exchange, and remineralisation of organic matter (Schneider

et al., 2014b). Beginning in September/October, deepening of the mixed layer leads to increasing $pCO_2$ in the surface water as a consequence of entrainment of deeper waters that have been subject to organic matter remineralisation (Schneider et al., 2014a). Typical $pCO_2$ values in the central Baltic Sea range from about 100 to 500 µatm (Schneider et al., 2014a).

The $CH_4$ distribution in the Baltic Sea is mainly governed by the vertical redox and density stratification (Fig. 2): Anoxic, $CH_4$-enriched bottom water in the deep basins and $CH_4$ oxidation in the water column, especially in the redox zone (Jakobs

et al., 2013), lead to strong vertical gradients and concentrations near atmospheric equilibrium in the surface water (Schmale et al., 2010; Jakobs et al., 2013). $CH_4$ is also produced in the upper, oxic water column (Jakobs et al., 2014; Schmale et al., 2018), partially by zooplankton (Schmale et al., 2018; Stawiarski et al., 2019). The Baltic Sea acts as a permanent source of atmospheric $CH_4$. Highest exports are observed in the Gulf of Finland in winter (Gülzow et al., 2013) and in the shallow western basins due to $CH_4$-rich sediments (Schmale et al., 2010).

Local coastal upwelling increases both $pCO_2$ and $CH_4$ concentration ($cCH_4$) in the surface water of the Baltic Sea by introducing enriched water from below the summer thermocline (Fig. 2) to the surface (Gülzow et al., 2013; Norman et al., 2013; Schneider et al., 2014b; Humborg et al., 2019; Stawiarski et al., 2019). However, since previous studies were limited to





episodic events, only little is known about the seasonality and regional distribution of upwelling-induced trace gas signals in the Baltic Sea and their relaxation. Ferry-based measurements enable studies on upwelling in the Baltic Sea on larger scales, which has already been demonstrated for physical parameters in the Gulf of Finland (Kikas and Lips, 2016).

Upwelling events in the Baltic Sea are common, but irregular since they depend on wind conditions (Lehmann and Myrberg, 2008). Westerly to south-westerly winds prevail in the Baltic Sea area, which enhances the possibility of upwelling near southern and south-eastern coasts (see red arrows in Fig. 1). These upwelling events have a typical lifetime of several days up to one month with sharper horizontal gradients compared to oceanic upwelling (Lehmann and Myrberg, 2008). In summer, sea surface temperature (SST) may decrease by more than 10 °C during an upwelling event, while salinity changes are usually below 0.5 (Lehmann and Myrberg, 2008). The influence of upwelling decreases in autumn and winter, when no seasonal stratification is present. While for oceanic upwelling regions, it is known that upwelling may trigger extreme primary production through nutrient transport, the influence of upwelling on primary production in the Baltic Sea is still poorly constrained (Lehmann and Myrberg, 2008). However, upwelled waters characterised by low N/P ratios have been reported to fuel cyanobacteria blooms during nitrogen limitation (Vahtera et al., 2005; Lips et al., 2009; Wasmund et al., 2012). Yet, time lags of about three weeks are possible for this feedback with an initial decline of phytoplankton biomass (Vahtera et al., 2005; Wasmund et al., 2012). As an explanation for this delay, Wasmund et al. (2012) propose that the initialisation of a cyanobacteria bloom requires mixing of biomass-rich surface water with phosphate-rich upwelled water.

The co-occurrence of episodic upwelling periods and dynamic trace gas patterns, and the combination of the presented eight-years data set of SOOP *Finnmaid* and of high-resolution models (Placke et al., 2018; Gräwe et al., 2019) allow us to investigate the influence of coastal upwelling on surface $p\mathrm{CO}_2$ and $c\mathrm{CH}_4$ in the Baltic Sea on a large spatial and temporal scale. In this study we are aiming at:

- presenting a method to identify upwelling events within the data set based on wind and modelled SST data,

- comparing upwelling-induced trace gas dynamics within several regions in the Baltic Sea,

- examining the relaxation of upwelling events over time with a focus on underlying processes,

- discussing interannual variability within the data set and highlighting the importance of upwelling to understand $CO_2$ and $CH_4$ dynamics in the Baltic Sea, and

- demonstrating the potential of extrapolating trace gas observations based on modelled SST data to estimate air–sea trace gas fluxes on a broader spatial scale.

We present most of the findings by use of illustrating examples, but provide more information in the appendix, supplement, and data set.



## 2 Data and methods

### 2.1 Measurements aboard SOOP *Finnmaid*

SOOP *Finnmaid* is equipped with a variety of sensors to survey the surface water of the Baltic Sea between Lübeck-Travemünde
in Germany and Helsinki in Finland (Fig. 1). Parameters including SST, salinity, $pCO_2$, and $cCH_4$ are logged every minute.
The data set used for this study is filtered from May to September and within the regions defined in Table 1 and consists of 482
transects from 2010 to 2017 with about 395,000 observations.

The on-board trace gas measurement system consists of a *Los Gatos Research* $CH_4/CO_2$ analyser coupled with an air–water
equilibrator (Körtzinger et al., 1996), which is described in detail in Schneider et al. (2014b) and Gülzow et al. (2011, 2013)
including details on the following calculations: The measured variables are the mole fractions $xCO_2$ and $xCH_4$ in ppm, which
are corrected to dry-air values using $xH_2O$ data from the same instrument. These mole fractions are converted into partial
pressures using atmospheric pressure data and calculating saturation water vapour pressure following Weiss and Price (1980)
under the assumption of 100 % humidity in the equilibrator headspace. In literature, $CO_2$ data are usually reported as partial
pressure or fugacity, while $CH_4$ data are converted into concentrations. Accordingly, we report $pCO_2$ and $cCH_4$, the latter
was calculated from the partial pressure ($pCH_4$) using Bunsen solubility coefficients determined by Wiesenburg and Guinasso
(1979). Note that concentration is a conservative parameter with respect to temperature changes, while partial pressure is
temperature-dependent (see also Sect. 3.3). To compensate the effect of water warming from inlet to equilibrator, $pCO_2$ is
temperature-corrected following Takahashi et al. (1993).

We post-calibrated $xCO_2$ and $xCH_4$ using a single-point calibration to a standard gas at near-atmospheric concentrations.
These standard gas measurements were performed automatically when leaving the harbour to yield a drift correction between
transects. The measurement system aboard SOOP *Finnmaid* also includes a *LI-COR* 6262 $CO_2/H_2O$ analyser with a sepa-
rate equilibrator. Even though these additional $CO_2$ data are not presented here, they provided cross-validation and quality
control. Presenting both $CO_2$ and $CH_4$ measurements from the same instrument ensures best consistency between the two
trace gases. Therefore, minor deviations from the previously published $CO_2$ data set in SOCAT (Surface Ocean $CO_2$ Atlas,
https://www.socat.info, Bakker et al., 2016) exist, which is a combined product of both setups. In the study area and period,
these differences in $pCO_2$ have a median of 0.75 µatm and an interquartile range (IQR) of 2.1 µatm.

We further used monthly-averaged atmospheric $CO_2$ and $CH_4$ data to calculate atmospheric equilibrium conditions. For the
closest distance to observations from SOOP *Finnmaid*, we utilised atmospheric data from Utö station (Finnish Meteorological
Institute, Helsinki) starting in March 2012. Prior to that or to fill gaps in the Utö series, atmospheric data from Mace Head
station (National University of Ireland, Galway) via the NOAA ESRL Carbon Cycle Cooperative Global Air Sampling Network
(Dlugokencky et al., 2019a, b) were used and both data sets were matched to those of Utö by linear regression (Fig. A1 for
details). The atmospheric data are displayed as atmospheric partial pressure for $CO_2$ or as saturation concentration calculated
from SST and salinity for $CH_4$. We also plotted relative $CH_4$ saturation, which is the ratio of $cCH_4$ to saturation concentration.

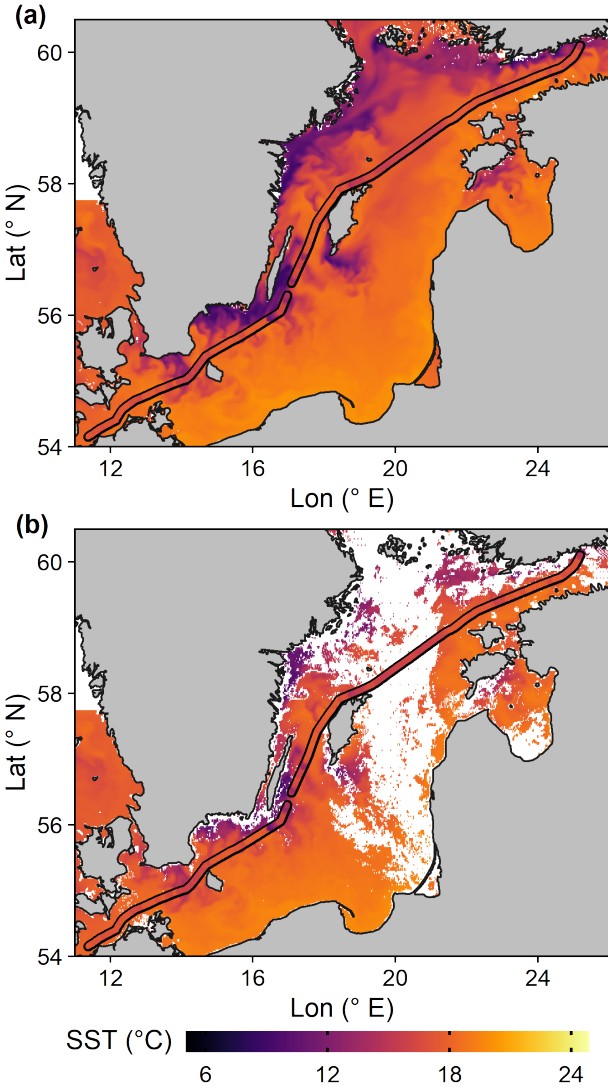

**Figure 3.** SST in the Baltic Sea on 16 August 2016 (daily mean) as extracted from (a) the model and (b) the remote sensing product (see text). Supplement S1 contains an animation over time with the same colour scale. SST measurements aboard SOOP *Finnmaid* from the same day are plotted on top of both panels. We chose this day for demonstration purposes as best compromise between remote sensing and SOOP data coverage as well as observable upwelling signals.

## 2.2 Wind and modelled SST data

Wind-induced upwelling in summer results in decreasing SST. Thus, to find upwelling-induced trace gas signals within the data set of SOOP *Finnmaid* and to assess the spatial and temporal dimensions of upwelling events, we combined reanalysed wind and modelled SST data to locate upwelling events in space and time before starting the actual in situ data analysis. Using





modelled SST data enabled us to also identify events that SOOP *Finnmaid* missed, without cloud-coverage limitations present in remote sensing data.

We used the SST output of the numerical ocean model GETM (General Estuarine Transport Model) for the Baltic Sea. The model has a horizontal resolution of 1 nautical mile and 50 vertical terrain-following levels. The uppermost level has a maximum thickness of 50 cm to properly represent the SST and ocean–atmosphere fluxes. The model run covers the period from 1961 to 2019. For a detailed analysis of the model performance see Placke et al. (2018) and Gräwe et al. (2019). For the present run, we restarted the model in 2003, but changed the atmospheric forcing to the operational reanalysis data set of

the German Weather Service (DWD), with a spatial resolution of 7 km and a temporal resolution of 3 h (Zängl et al., 2015). The same wind data – extracted for one location per upwelling region, respectively (Fig. 1) – were used to identify upwelling events (Sect. 2.3). To give an impression of the model performance, Fig. 3 and the Supplement S1 illustrate SST in the Baltic Sea during a strong upwelling event taken from both the model and a multi-sensor level 3 SST remote sensing product for the European seas (Copernicus, 2020). This comparison demonstrates both good agreement and the advantage of model data due

to insensitivity to cloud coverage. Differences between modelled SST data along the track of SOOP *Finnmaid* and shipboard SST observations in the entire study area and period have a median of 0.04 °C and an IQR of 1.41 °C. Differences between model and observations partly result from different time scales, i.e. daily means (model data) vs. real time (in situ data).

## 2.3  Identification of upwelling events

Based on the statistical analysis of upwelling in the Baltic Sea by Lehmann et al. (2012), we defined major upwelling areas

that SOOP *Finnmaid* crosses as foci of this study (Fig. 1 and Table 1). We excluded the Arkona Basin and the Mecklenburg Bight (areas west of 14° E) because strong wind may trigger vertical mixing through the entire water column in these shallower areas, thereby eliminating the usual decoupling between sediment and surface water and greatly enhancing surface trace gas concentrations (Gülzow et al., 2013). Thus, it is impossible to disentangle the influence of wind-induced upwelling in these areas by the method proposed here. We included an area in the open Gotland Sea, which should not be directly influenced by

upwelling due to being far from the coast (> 40 km, Table 1), for comparison. Furthermore, we only considered data from May to September each year, when upwelling-induced SST signals can be observed (Lehmann et al., 2012), which is – together with a wind criterion – the basis of the detection method we used.

According to Lehmann et al. (2012), we defined an upwelling event as: upwelling-favourable wind, i.e. the wind component projected parallel to the coast (Fig. 1) exceeding 3.5 ms$^{-1}$ for two days, causing a temperature drop by more than 2 °C in

the respective box. Both criteria (wind and ΔSST) were evaluated per day and box and visualised in yearly plots, which also display data coverage of SOOP *Finnmaid* (Fig. 4, A2, and A3).

We calculated the wind criterion as running mean of upwelling-favourable wind speeds of the last 48 h with a temporal resolution of 3 h. The criterion is considered as "met" if at least 4 of the 8 mean values per day exceed the threshold of 3.5 ms$^{-1}$.

The ΔSST criterion was calculated as difference between median and minimum model-SST in the respective area, since a local upwelling event will lower the minimum SST while the median remains relatively stable. To achieve a more robust



**Table 1.** Upwelling areas crossed by SOOP *Finnmaid* (abbreviated and long name), their boundaries (latitude/longitude) including a specification if the respective box is not rectangular (Fig. 1), their upwelling-favourable wind direction and the distance between the track of SOOP *Finnmaid* within the box and the coast given as median and minimum. This distance is calculated perpendicular to the upwelling-favourable wind direction and based on an average track.

| Abbrev. | Long name | Lat (° N) | Lon (° E) | Specification | Upw.-fav. wind direction (°) | Med. (min.) dist. to coast (km) |
|---|---|---|---|---|---|---|
| Born | close to Bornholm | 54.75–55.75 | 14–15 | – | 225 (45)[a] | 43 (21) |
| Ö-S | S of Öland | 55.5–56.2 | 15.5–17 | – | 260 | 42 (25) |
| Ö-E | E of Öland | 56.2–57.5 | 16.5–17.5 | only E of Öland | 210 | 27 (21) |
| Go-SE | SE of Gotland | 56.5–57.75 | 18–19.5 | only SE of Gotland | 235 | 25 (17) |
| Go-NW | NW of Gotland | 57.25–58.25 | 17.8–19.5 | only NW of Gotland | 65 | 10 (4) |
| Hiiu | close to Hiiumaa | 58.75–59.5 | 21.5–22.5 | – | 75 | 43 (27) |
| GoF | Gulf of Finland | 59.25–60 | 23–24.5 | only SE of Hanko Peninsula | 260 (80)[a] | 28 (19) |
| openGo | open Gotland Sea | 58–59 | 20–21 | – | none[b] | 64 (40)[c] |

[a] Wind directions in brackets have been considered but found to have only little influence on data from SOOP *Finnmaid*.

[b] Due to large distances to surrounding coasts.

[c] Calculated as nearest coast in any direction.

median, the boxes were selected to extend beyond the actual upwelling areas. This results in a pronounced increase in ΔSST during upwelling events, while the criterion is mostly below the 2 °C threshold otherwise. This calculation can be based on either, first, the entire area inside the boxes (Fig. A2) or just, second, on sub-transects, i.e along the track of SOOP *Finnmaid*
(Fig. 4). We mainly used the latter for the purposes of this study, which is justified by a comparison of the capability of both approaches to match a daily ΔSST criterion calculated from SST observations aboard SOOP *Finnmaid*: The second approach based on sub-transects is less sensitive (hit rate: 0.57 vs. 0.94 for the first approach), but has a higher specificity (false alarm rate: 0.08 vs. 0.48) and better skill to forecast correctly (proportion correct: 0.88 vs. 0.57; critical success index: 0.36 vs. 0.21). These verification measures were calculated according to Jolliffe and Stephenson (2003). The differences in number of events
correctly identified depending on which subset of SST data is used are explained by the fact that upwelling events start near the coast and then propagate seawards and, thus, SST drops along the sub-transect are delayed and often smaller. The first approach (using all SST data within the upwelling box) does not incorporate this lag, but is usually better aligned with the wind criterion for the same reason. Therefore, the appropriate method choice depends on the desired use: Including more spatial coverage of SST data would be appropriate to analyse the occurrence of upwelling in a certain region statistically. However, we chose
to use only SST data along the SOOP route to amplify the agreement with the in situ SST and trace gas measurements. We provide a more detailed method assessment in Sect. 3.1.


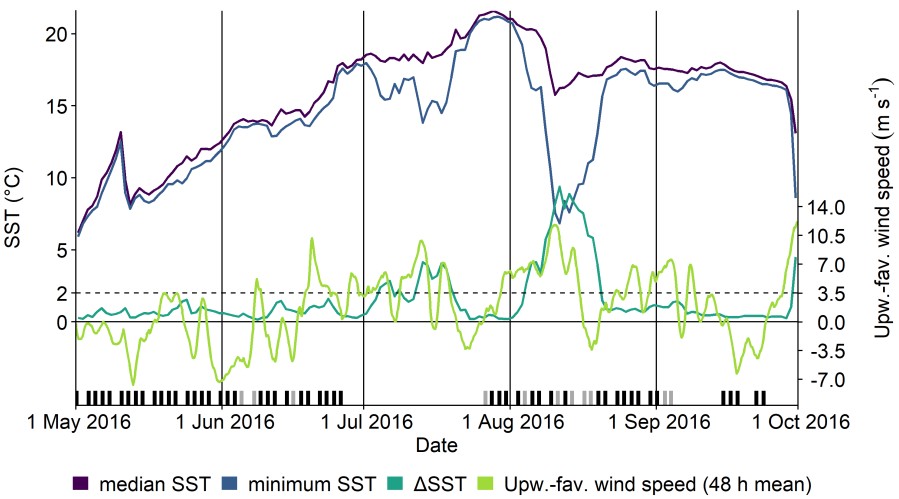

**Figure 4.** Time series to demonstrate upwelling detection within box Go-SE. Purple and blue lines are median and minimum model-SST along the transect of SOOP *Finnmaid*; the turquoise line shows their difference ($\Delta$SST). The green line represents the running mean of upwelling-favourable wind speeds, calculated every 3 h for the last 48 h, respectively. The dashed black line indicates the chosen thresholds of the $\Delta$SST and wind criteria, respectively (2 °C, 3.5 m s$^{-1}$). Each passage of SOOP *Finnmaid* through the box is marked with a black dash at the bottom. Grey dashes mark when the ship took the western route around Gotland, thereby missing this particular box on the east side. A strong upwelling event in August is observed with small data gaps due to SOOP *Finnmaid* taking the western route. Another event in July is missed because of a sensor malfunction.

Note that a large-scale upwelling event triggers a drop in median SST, but due to increased spatial variability during those events, the sensitivity of $\Delta$SST is usually still sufficient to exceed the threshold of 2 °C in these cases (e.g. Fig. 4 and A2 on ca. 10 August 2016).

## 2.4 Calculation of theoretical relaxation and flux estimates

We calculated theoretical relaxation curves (Sect. 3.3) as follows: CO$_2$ system calculations were performed using the R package *seacarb* (Gattuso et al., 2019) with K$_1$/K$_2$ from Millero (2010), K$_w$/K$_f$ from Dickson and Riley (1979) and K$_S$ from Dickson (1990). $p$CH$_4$ was calculated from $c$CH$_4$ (Wiesenburg and Guinasso, 1979). Since salinity changes by upwelling in the Baltic Sea are usually small (Lehmann and Myrberg, 2008) and no calcifying organisms are present (Schneider et al., 2014a), we

assumed a constant salinity of 7 and a total alkalinity (A$_T$) of 1600 µmol kg$^{-1}$ (Müller et al., 2016). These along with the values for the initial (upwelled) water mass (SST = 10 °C, $p$CO$_2$ = 700 µatm, $c$CH$_4$ = 5 nmol L$^{-1}$) and the background water mass (SST = 20 °C, $p$CO$_2$ = 150 µatm, $c$CH$_4$ = 3.5 nmol L$^{-1}$) used for mixing are typical for late-summer upwelling in the central box Go-SE (Fig. 8d,l). However, as we mainly discuss the shapes of the curves, which are unaffected by variation of the input variables over a reasonable range, Fig. 9 is used to discuss processes in all regions.


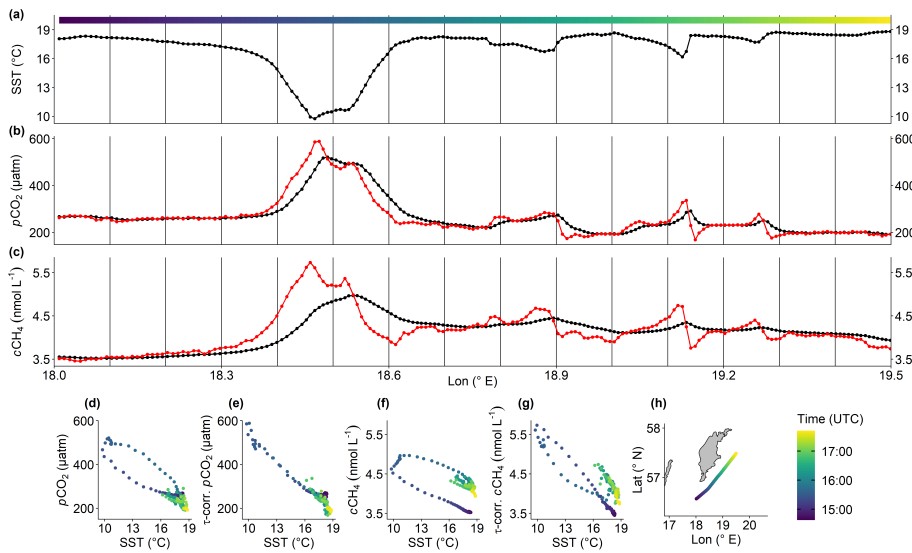

**Figure 5.** Demonstration of response time correction based on data from 24 August 2010 within box Go-SE. The three upper plots display longitudinal patterns of (a) SST, (b) $p\mathrm{CO_2}$, and (c) $c\mathrm{CH_4}$. Black symbols are original values, red symbols are corrected using response times of $\tau_{\mathrm{CO_2}}$ = 226 s and $\tau_{\mathrm{CH_4}}$ = 676 s (Gülzow et al., 2011) and procedures according to Bittig et al. (2018). (d–g) reveal relationships between (d) original and (e) $\tau$-corrected $p\mathrm{CO_2}$ and SST, as well as (f) original and (g) $\tau$-corrected $c\mathrm{CH_4}$ and SST. (h) displays the position of SOOP *Finnmaid* over time. Time is colour-coded to link all plots.

185       Fluxes were calculated according to Wanninkhof (2014). We approximated the Schmidt number dependence on salinity via linear interpolation between the values for freshwater and seawater. In Sect. 3.3, we assumed constant wind speeds of $10\,\mathrm{ms^{-1}}$, SST = $10\,^\circ\mathrm{C}$, air temperature = $20\,^\circ\mathrm{C}$, relative humidity = 0.8, and relative cloud coverage = 0.8. In Sect. 3.5, we used the available wind data with 3 h resolution to calculate daily fluxes.

## 2.5   Air–water equilibrator response times

Gas phase measurements using air–water equilibrators are subject to response times (Johnson, 1999), which depend on construction and operation parameters of the setup, solubility of the respective gas, temperature and salinity (Webb et al., 2016). The *e*-folding time constants $\tau$ of the system aboard SOOP *Finnmaid* were determined to be 226 s for $\mathrm{CO_2}$ and 676 s for $\mathrm{CH_4}$ at room temperature using fresh water (Gülzow et al., 2011). Non-negligible response times lead to smoothed and delayed signals in both time and space, with a more pronounced impact on $\mathrm{CH_4}$ than $\mathrm{CO_2}$. Corrections for temporal and spatial lag
are used in profiling sensor applications (Fiedler et al., 2012; Bittig et al., 2014), but they are often neglected for surface trace gas measurements. We demonstrate such a correction using the method described in Bittig et al. (2018) to illustrate potential advantages and practical issues (Fig. 5).

      In the illustrated example, SOOP *Finnmaid* travels from south-west to north-east through box Go-SE (Fig. 5h). Compared to SST measurements (Fig. 5a), the $\mathrm{CO_2}$ and especially $\mathrm{CH_4}$ signals are delayed and smoothed (Fig. 5b,c, black curves). In com-





parison, the corrected signals (Fig. 5b,c, red curves, most prominently around 18.5° E) respond earlier, are more pronounced, exhibit more fine structure, and mirror the SST signal better, which is expected when entering a new water mass. The relationships between uncorrected trace gas signals and SST (Fig. 5d,f) feature hysteresis, which is reduced substantially after the correction (Fig. 5e,g). However, the method introduces artefacts like overshoots (e.g. Fig. 5b,c, low values around 19.15° E) and noise particularly if data density is low and/or $\tau$ is poorly characterised. This problem can be mitigated, but not solved, by

applying additional smoothing before and/or after the correction (not done here).

Unfortunately, this response time correction only provided satisfactory results for a minority of cases. Elsewhere, the resulting noise degraded data quality and created additional hysteresis. We attribute this to the unknown dependence of $\tau$ on, e.g. temperature, salinity, and water/gas flows, all of which vary along a transect. This issue would be particularly influential for this study since, e.g. SST gradients caused by upwelling are sharper and steeper compared to measurements in open basins, which

leads to perpetual changes of $\tau$. Thus, we decided to refrain from a response time correction to avoid introducing additional bias into the data set. However, the algorithm we used (Bittig et al., 2018) is capable of handling variable $\tau$, allowing more precise response time corrections if $\tau$ is sufficiently characterised as function of, e.g. temperature, salinity, and air/water flows, which leaves room for future studies.

## 2.6  Data processing and visualisation

Data analysis and visualisation were executed using R (R Core Team, 2019), particularly the packages *tidyverse* (Wickham et al., 2019), *cowplot* (Wilke, 2019), and colour scales from *viridis* (Garnier, 2018). For maps, we used bathymetry data from *marmap* (Pante and Simon-Bouhet, 2013) and coastline data from *rnaturalearth* (South, 2017). Three-dimensional plots were rendered with *rayshader* (Morgan-Wall, 2020).

## 3  Results and discussion

### 3.1  Upwelling statistics based on wind and modelled SST data

To assess the prevalence of upwelling in the data set of SOOP *Finnmaid*, we identified the main upwelling periods and areas along the transect using the method of combining a ΔSST and a wind criterion (Sect. 2.3). Here, we present a summary of the climatological mean number of days per month and box where the criteria were met (Fig. 6). The ΔSST criterion was calculated based on sub-transects. We provide a full overview further distinguishing by year and selection of SST data (entire

box vs. sub-transect) in the appendix (Fig. B1, B2, and B3). The wind criterion is usually met more frequently than the ΔSST criterion calculated along sub-transects. This reflects that not every occurrence of wind strong enough to induce upwelling leads to upwelled water masses actually reaching the track of SOOP *Finnmaid*. This is illustrated in Fig. 4 (June 2016) and in the Supplement S1. Downwelling may also lead to quickly vanishing signals (Sect. 3.3). In general, the ΔSST criterion is not very sensitive in May due to a less pronounced thermocline compared to summer. Similarly, only small upwelling-induced trace gas

signals are observed in May, which become greater in late summer owing to longer decoupling of surface water and underlying



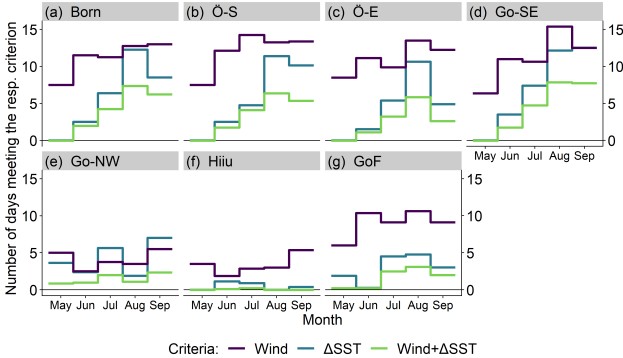

**Figure 6.** Overview of the two upwelling criteria used in this study: wind (purple) and $\Delta$SST (blue). Displayed is the number of days per month and box in which the respective criterion is met, averaged over 2010 to 2017. $\Delta$SST was calculated along the route of SOOP *Finnmaid* through the boxes (Sect. 2.3, second approach) to display events that are actually observable from the ship. "Wind+$\Delta$SST" (green) only applies to instances of both criteria being met on the exact same day and therefore excludes occasional instances of lag between wind and $\Delta$SST signals. Box openGo is not included here since no upwelling-favourable wind direction can be defined.

sub-thermocline waters (Sect. 3.4). Upwelling in autumn and winter either leads to a general deepening of the mixed-layer depth (discussed in Gülzow et al., 2013) or plays no important role when the physical and biogeochemical differences between surface and upwelled waters have vanished in winter.

The $\Delta$SST criterion based on the entire area is more sensitive than that calculated along sub-transects, but less specific

regarding the prediction of upwelling in dynamic areas like GoF since it is essentially a measure for SST variability $> 2\ ^{\circ}$C within the box (Sect. 2.3). It is triggered more frequently than the wind criterion and due to the high sensitivity of $\Delta$SST, the agreement with the wind criterion and, thus, both criteria being met, is high (Fig. B3).

Boxes Born, Ö-S, Ö-E, Go-SE, and GoF follow similar patterns with respect to both criteria (Fig. 6), which is not surprising given the fact that in all of these cases, upwelling is induced by the same south-westerly to westerly winds and the minimum

distances to the coast are rather similar. The upwelling-favourable wind direction is opposite in boxes Go-NW and Hiiu. In box Hiiu, however, both criteria are almost never met at the same time, which indicates that the distance between sub-transect and coast is too large to observe strong upwelling signals (minimum 27 km, median 43 km, Table 1). Admittedly, the sub-transect in box Ö-S is comparable to Hiiu in terms of distance to the coast, but the crucial difference seems to be the upwelling-favourable wind direction since strong westerly winds are more frequent and intense. This is supported by the fact that, even if we calculate

$\Delta$SST based on the entire area, the number of days where both criteria are met in box Ö-S is higher than in box Hiiu (Fig. B3), clearly indicating that upwelling is more common in Ö-S. In contrast, the sub-transect in box Go-NW is frequently influenced by upwelling, but yet, it is the only box where the $\Delta$SST is met more often than the wind criterion (Fig. 6e). We attribute this to the small distance to the coast (minimum 4 km, median 10 km, Table 1) leading to higher SST variability and, thus, more similarity to the $\Delta$SST criterion that includes the entire area (Fig. B3e).





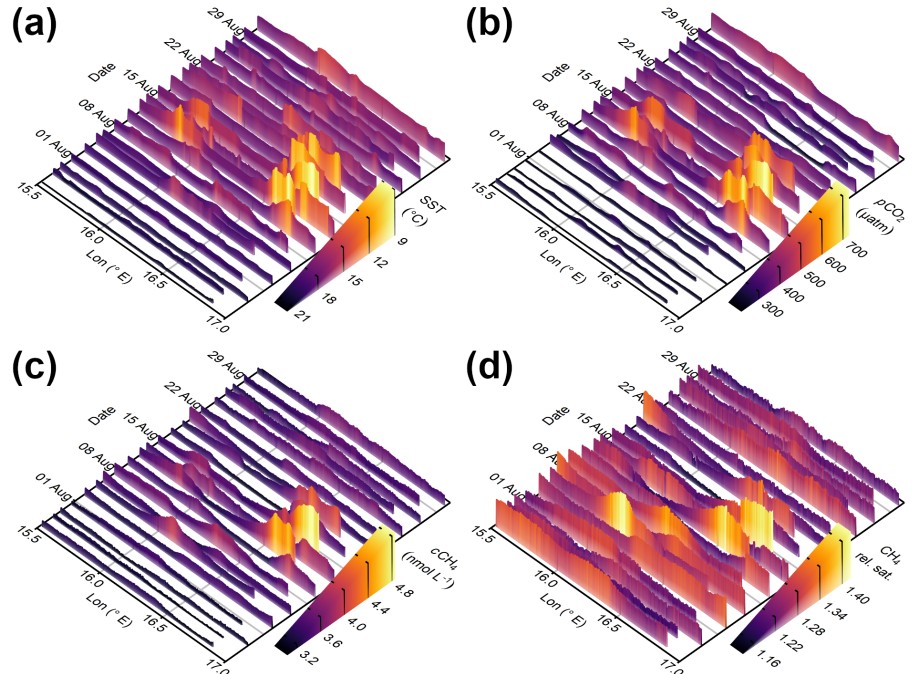

**Figure 7.** (a) Sea surface temperature, (b) $CO_2$ partial pressure, (c) $CH_4$ concentration, and (d) relative $CH_4$ saturation within box Ö-S as measured by SOOP *Finnmaid* on 21 sub-transects from 26 July to 30 August 2016. In all panels, abscissa is position given as longitude, ordinate is time, and the respective variable is displayed by both colour and height of the curve. Please note the inverted SST scale in (a) to highlight the correlation between decreasing SST and increasing $p$CO$_2$ and $c$CH$_4$. Data presented in (a–c) correspond to Fig. 8b,j.

Based on this statistical analysis, we chose box Ö-S as example area for most of the following discussion since it features prominent upwelling signals concerning both frequency and magnitude. Additionally, data coverage in this area is high as it is crossed by SOOP *Finnmaid* on either of its routes (Fig. 1).

## 3.2   Regional comparison of upwelling events

In this section, we investigate upwelling events that were caused by strong winds across the entire study area in August 2016,
leading to temperature and trace gas signals in almost all previously defined upwelling areas (Fig. B1 and B2), which allows us to compare the observations in these regions and to assess the importance of upwelling for the observed trace gas dynamics. This case study exemplifies more general findings we gained during the analysis of the entire data set.

The entire month was characterised by strong westerly to south-westerly winds (Fig. 4), leading to upwelling in boxes Born, Ö-S, Ö-E, Go-SE, and GoF, interrupted by a week (15–22 August 2016) of more north-easterly winds, triggering upwelling in
boxes Hiiu and Go-NW. The resulting SST drops by up to 16 °C predominantly near all southern and eastern coasts propagated seawards and relaxed within several weeks (Supplement S1). Coverage of data from SOOP *Finnmaid* in this period is very





dense, with the majority of transects along the east side of Gotland (see ratio of black and grey dashes in Fig. 4). We illustrate the temporal and spatial evolution of this event (Fig. 7) taking the example of box Ö-S.

Before the event, SST is at a typical summer value of 21 °C throughout the entire sub-transect. As expected, upwelling leads to SST decreases (displayed as peaks in Fig. 7a) with temperatures down to 9 °C. These minima move over time (see also Supplement S1) and are subject to relaxation, which is further discussed in Sect. 3.3. The pre-upwelling temperature is usually not reached again during this time of the year, most probably due to an increased mixed-layer depth as an additional effect of stronger winds, and weakened solar irradiation at the end of August.

The observed trace gas patterns are similar to the temperature distribution: The first sub-transect can be considered as background conditions with typical late summer values of about 250 $\mu atm$ for $pCO_2$ and 3.2 $nmol\,L^{-1}$ for $cCH_4$ in this area. During the upwelling event, we observe elevated $pCO_2$ and $cCH_4$, with trace gas maxima correlated to minimum SST. For $CO_2$, this results in a switch from undersaturation to supersaturation. $CH_4$ is always supersaturated or in equilibrium with the atmosphere in the SOOP *Finnmaid* data set and strong upwelling further increases this supersaturation and, eventually, $CH_4$ outgassing. However, upwelling is not the only factor controlling increased $CH_4$ supersaturation (Fig. 7d). Warming of upwelled waters increases $pCH_4$ and, therefore, relative saturation. As with SST, the enhanced trace gas levels relax subsequentially.

To extend these findings to the different regions, we investigated the relationships between trace gas data and temperature over time (Fig. 8). The example from box Ö-S is representative of the majority of strong upwelling events affecting trace gases in the data set of SOOP *Finnmaid*, which are generally characterised by near-linear relationships between trace gases and SST. Maximum $pCO_2$ and $cCH_4$ values would not be reached without upwelling in summer (Fig. B4). We observe the same behaviour in boxes Go-SE, Ö-E, and Go-NW despite their reduced data coverage (Fig. 8). Box Ö-E features the highest $pCO_2$ in the data set of over 800 $\mu atm$. In box Go-NW, the observable upwelling event began only at the end of trace gas data coverage on the western route due to a different favourable wind direction, hence, the more extreme values are missing in this example. However, the resulting pattern resembles the ones of boxes Ö-S, Ö-E, and Go-SE, just with lower maximum $pCO_2$ and $cCH_4$.

Boxes Born and GoF show the same relationships as the previous boxes in their $pCO_2$–SST diagram with considerable dynamic range in case of box GoF (Fig. 8a,g). The respective $cCH_4$–SST diagrams, however, only contain a small branch of increasing $cCH_4$ with decreasing SST (Fig. 8i,o). These regions are dominated by temperature-independent $CH_4$ variability, which indicates that other processes than upwelling might cause higher-than-usual $cCH_4$: Box Born is situated between two basins (Arkona and Bornholm basin), which are interlinked via lateral transport, and which both feature gassy sediments (Gülzow et al., 2014; Tóth et al., 2014), from which $CH_4$ may be released via pressure changes caused by strong winds (Schneider von Deimling et al., 2010; Gülzow et al., 2013). $cCH_4$ variability in box GoF might be driven by the highly variable physical conditions, e.g. changes of the estuarine circulation up to full reversal (Westerlund et al., 2019) or enhanced vertical transport by boundary wall shear (Schmale et al., 2010). These effects would lead to a less distinct impact of upwelling compared to boxes Ö-S, Ö-E, Go-SE, and Go-NW, where vertical decoupling is more stable.

The discussed phenomena can be contrasted with the behaviour of the sub-transect within box Hiiu: Although we find a clear correlation between decreasing SST and increasing $pCO_2$ resembling that of the other boxes, we do not observe temperatures



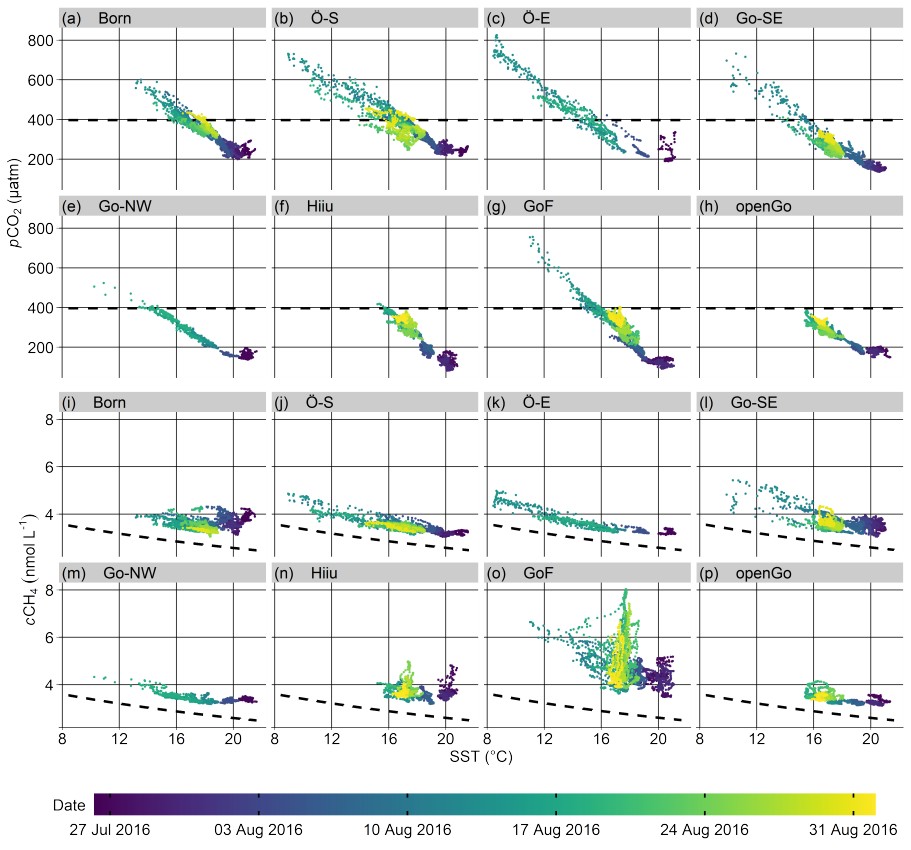

**Figure 8.** Surface $p\mathrm{CO_2}$ (a–h) and $c\mathrm{CH_4}$ (i–p) as measured by SOOP *Finnmaid* on 21 transects from 26 July to 30 August 2016, each plotted against SST within the seven upwelling regions and the open Gotland Sea box for comparison. The measurement date is colour-coded. Temporal coverage in box Go-SE (15 transects, see black dashes in Fig. 4) and boxes Ö-E and Go-NW (6 transects, see grey dashes therein) is reduced since SOOP *Finnmaid* uses two different routes around Gotland. Black dashed lines indicate atmospheric equilibrium partial pressure for $\mathrm{CO_2}$ and concentration for $\mathrm{CH_4}$ (calculated using mean salinity per box in the given time period), respectively.

lower than 15 °C and $p\mathrm{CO_2}$ higher than 420 µatm, which is close to atmospheric equilibrium (Fig. 8f). This relationship is similar to that of box openGo, the patterns in which we attribute to mixed-layer deepening and air–sea gas exchange caused by stronger winds instead of upwelling because of its distance to the coast (minimum 40 km, median 64 km, Table 1). Since the route of SOOP *Finnmaid* within box Hiiu is the furthest away from the coast of all boxes (minimum 27 km, median 43 km, Table 1) and the observed $p\mathrm{CO_2}$–SST relationships are so similar to openGo, we infer that upwelling has only minor influence on the observed values in this region during this time period. This is consistent with maps of modelled SST (Fig. 3 and Supplement S1, most pronounced around 18 August 2016), where no upwelled water masses reach out to the ship track, and confirms the same finding from the statistical identification of main upwelling areas presented in Sect. 3.1.






The $c$CH$_4$–SST diagram of box Hiiu (Fig. 8n) resembles, to some extent, that of box GoF (Fig. 8o) without the upwelling branch and smaller maximum $c$CH$_4$, and indicates considerable $c$CH$_4$ variability compared to, e.g. boxes Ö-S, Ö-E, Go-SE, and Go-NW. In late July, for example, CH$_4$ concentration drops from 4.8 to 3.3 $\mathrm{nmol\,L^{-1}}$ at more or less constant temperature (Fig. 8n), equating to a change in relative CH$_4$ saturation from 1.9 to 1.3. In the adjacent region openGo, no instances of increasing $c$CH$_4$ at constant SST (vertical branches in Fig. 8n–p) were observed.

We summarise that upwelling affects observed SST, $p$CO$_2$, and $c$CH$_4$ drastically in the defined boxes in late summer of 2016. It typically causes near-linear relationships between surface trace gas signals and temperature with varying ranges and slopes between regions. For CO$_2$, this can be observed in all regions (with limitations in box Hiiu), while in the case of CH$_4$, strong variability caused by other processes may mask the effects of upwelling and closest-to-linear relationships are observed in boxes Ö-S, Ö-E, Go-SE, and Go-NW.

## 315 3.3 Typical relaxation of upwelling-induced trace gas signals

The surface water properties of a region influenced by upwelling change over the course of the upwelling event. This can be seen in Fig. 8, where the evolution of the relationship over time between SST and $p$CO$_2$ or $c$CH$_4$ is indicated by colour, and in Fig. B4. Before an event, we observe high temperature and low trace gas levels, with low spatial variability within sub-transects. During strong upwelling, a variety of SST, $p$CO$_2$, and $c$CH$_4$ are observed concurrently as SOOP *Finnmaid*

transects the respective region. These values usually form a near-linear relationship. After upwelling-favourable winds cease, the range of signals as well as their intensity is reduced through relaxation in a quasi-linear fashion (Fig. 8). The final state after relaxation, when compared to the initial state, is shifted towards lower temperatures, higher $p$CO$_2$, and slightly elevated $c$CH$_4$, which equals a roughly comparable CH$_4$ supersaturation at this decreased, final temperature. Depending on time of the year, SST might re-increase due to subsequent warming, or not recover completely (as in late summer).

In order to discuss the processes that are involved in the relaxation of upwelling signals, we calculated theoretical relaxation curves in trace gas – temperature diagrams (Fig. 9). Assumed endmember characteristics, physical driving parameters, and process descriptions are summarised in Sect. 2.4. We focus on air–sea gas exchange, air–sea heat exchange, and mixing with a typical water mass with pre-upwelling conditions. CH$_4$ oxidation in the upper, oxic water column should not play a major role on the short time scales considered here (Jakobs et al., 2013). Primary production (e.g. by nitrogen fixation) has the potential

to decrease $p$CO$_2$ distinctly, but is difficult to constrain since it depends on meteorological conditions and nutrient availability with possible time lags of several weeks (Vahtera et al., 2005; Wasmund et al., 2012).

The relaxation of SST is mainly driven by mixing. We estimated a total surface heat flux of ca. 300 $\mathrm{J\,m^{-2}\,s^{-1}}$, which translates to a daily SST change of ca. 0.4 $\mathrm{K\,d^{-1}}$ assuming a mixed-layer depth of 15 m, which is rather typical for windy conditions in summer (derived from model data, not shown). Therefore, air–sea heat exchange contributes only little to the

observed warming of upwelled water masses, leaving mixing as the dominant process. Despite the excess of the surrounding water masses, mixing does not necessarily lead to pre-upwelling conditions since the endmember may change due to enhanced mixing in the open basins caused by stronger wind. SST might re-increase in the following weeks depending on meteorological conditions.





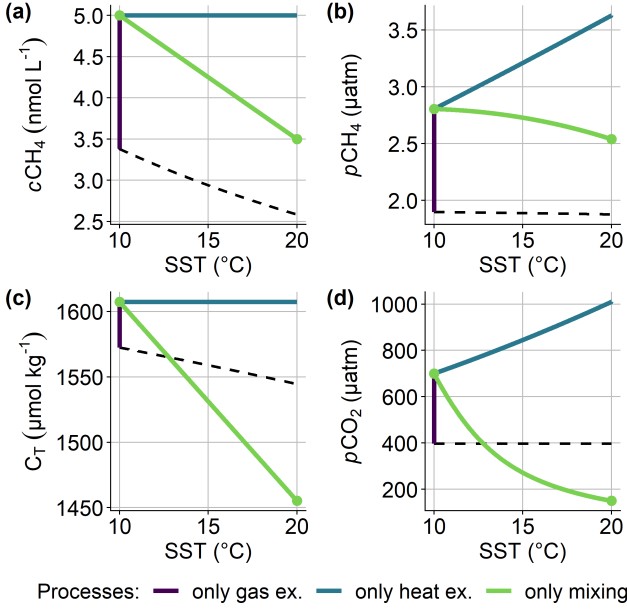

Processes: — only gas ex. — only heat ex. — only mixing

**Figure 9.** Theoretical relaxation curves of surface trace gas and temperature signals caused by upwelling. We calculated all graphs based on the processes (i) air–sea gas exchange, (ii) air–sea heat exchange, and (iii) mixing with a typical water mass with pre-upwelling conditions, with only one process considered at a time, starting at the point where the three lines intersect. Bold green points highlight the mixing endmembers. Black, dashed lines indicate atmospheric equilibrium conditions for the respective trace gas (397 ppm of $CO_2$, 1920 ppb of $CH_4$). Mixing lines were calculated for (a) $cCH_4$ using linear interpolation between endmembers (conservative behaviour), (b) $pCH_4$ from $cCH_4$ and SST, (c) $C_T$ using linear interpolation between endmembers whose $C_T$ was calculated from $A_T$ and $pCO_2$ (conservative behaviour), and (d) $pCO_2$ from $C_T$ and $A_T$. See Sect. 2.4 for details concerning calculation parameters.

Mixing also shapes the typically observed $cCH_4$–SST relationships (Fig. 8i–o and 9a), leading to near-linear mixing curves
since concentration is a conservative parameter with respect to temperature changes (we neglect the influence on water volume). The upwelled water mass releases ca. 7700 $\mathrm{nmol\,m^{-2}\,d^{-1}}$ of $CH_4$ into the atmosphere. This results in a daily $cCH_4$ loss of 0.51 $\mathrm{nmol\,L^{-1}\,d^{-1}}$ in a 15 m mixed layer, which is an efficient sink considering the magnitude of observed concentrations. Therefore, air–sea gas exchange alters the slope of the $cCH_4$–SST relationship. Note, however, that gas flux is highly dependent on wind speeds, which are biased in $cCH_4$–SST diagrams presented here: Pre-upwelling conditions involve low wind speeds,
while the upwelling event is caused by stronger winds, which eventually weaken. Heat exchange has no influence on $cCH_4$, but the relative $CH_4$ saturation is determined by its partial pressure $pCH_4$, which increases by the order of 2 % $\mathrm{K^{-1}}$ (Wiesenburg and Guinasso, 1979). This effect should not play a major role concerning relaxation given the low surface heat flux. However, as outlined above, SST might re-increase in the following weeks, thereby increasing $pCH_4$ and, thus, potentially lead to enhanced fluxes into the atmosphere. Likewise, mixing leads to elevated $pCH_4$ and relative saturation compared to linear behaviour
(Fig. 9b).



Similarly, the relaxation of $p\mathrm{CO_2}$ cannot be considered independently from SST relaxation due to its temperature dependence. Warming by air–sea heat exchange causes a $p\mathrm{CO_2}$ increase in the order of $4\ \%\,\mathrm{K^{-1}}$ (Takahashi et al., 1993), which should not play a major role concerning relaxation given the low surface heat flux. As with $p\mathrm{CH_4}$, however, this effect could lead to increasing $p\mathrm{CO_2}$ and enhanced $\mathrm{CO_2}$ fluxes into the atmosphere (or reduced fluxes into the sea) in the following weeks.

The relaxation of upwelling-induced $p\mathrm{CO_2}$ signals (Fig. 8a–g) cannot be explained solely by mixing because the theoretical $p\mathrm{CO_2}$–SST mixing curve obtained from $\mathrm{CO_2}$ system calculations features a distinct curvature with lower $p\mathrm{CO_2}$ compared to linear behaviour (Fig. 9d). The observed near-linear relationship is likely caused by air–sea $\mathrm{CO_2}$ exchange: At a wind speed of $10\ \mathrm{m\,s^{-1}}$, the upwelled water mass in this example (Fig. 9d) releases ca. $0.074\ \mathrm{mol\,m^{-2}\,d^{-1}}$ of $\mathrm{CO_2}$ into the atmosphere, which translates into a daily $\mathrm{C_T}$ (total dissolved inorganic carbon) loss of $4.8\ \mathrm{\mu mol\,kg^{-1}\,d^{-1}}$ in a $15\ \mathrm{m}$ mixed layer. This

$\mathrm{C_T}$ decrease in $\mathrm{CO_2}$-oversaturated waters explains the deviation from the expected linear (conservative) mixing curve in $\mathrm{C_T}$ estimated from $p\mathrm{CO_2}$ observations (Fig. B5 vs. 9c). Primary production triggered by upwelling has a similar (potentially even greater) influence on $p\mathrm{CO_2}$, but with different kinetics. One could argue that air–sea $\mathrm{CO_2}$ exchange should similarly increase $\mathrm{C_T}$ in $\mathrm{CO_2}$-undersaturated waters, which is not observed (Fig. B5). This can be explained with the aforementioned wind speed bias, resulting in very low fluxes under pre-upwelling conditions (see also Sect. 3.5). The bent $\mathrm{C_T}$–SST curve translates into

a near-linear $p\mathrm{CO_2}$–SST curve, which, in conclusion, can be interpreted as the combined result of mixing and decrease of the highest $p\mathrm{CO_2}$ values due to gas exchange and possibly primary production.

The importance of air–sea gas exchange as a relaxation process and the potential long-term effect of increasing supersaturation due to heat exchange imply that upwelling amplifies surface trace gas fluxes, especially for $\mathrm{CH_4}$ by circumventing the sink of $\mathrm{CH_4}$ oxidation in the water column. Schneider et al. (2014b) mentioned these upwelling-induced trace gas fluxes previously

for the Baltic Sea, but questioned the importance for the annual balance since the upper water column would be ventilated in autumn and winter anyhow and $\mathrm{CH_4}$ turnover times in the upper, oxic water column are in the magnitude of years (Jakobs et al., 2013). Despite a more detailed analysis of the statistical prevalence of upwelling in this study, the question of the importance of upwelling on the annual trace gas balance of the Baltic Sea still needs further investigation.

Another possible relaxation pathway is downwelling. Figure B6 provides an example of quickly vanishing upwelling signals

after turning wind. There, we expect downwelling to quickly remove upwelled waters from the surface, thereby restoring the previous surface water mass that underwent only small changes in SST, $p\mathrm{CO_2}$, and $c\mathrm{CH_4}$. This limits enhanced trace gas fluxes to a short time period during the upwelling event. This example is rather unique because it requires upwelling- and downwelling-favourable wind conditions in quick succession and can only be observed in close proximity to the coast (box Go-NW in this example), i.e. where the upwelled water mass is young and has not yet expanded towards the open sea. We can

exclude lateral transport out of the box as possible explanation for this effect based on maps of modelled SST (data not shown).

### 3.4    Interannual variability of upwelling-induced trace gas signals

Since upwelling in the Baltic Sea is an episodic phenomenon based on wind conditions, it is subject to considerable interannual variability. In Fig. 10, we present seasonal plots (from May to September) of $p\mathrm{CO_2}$ and $c\mathrm{CH_4}$ versus SST in box Ö-S. The

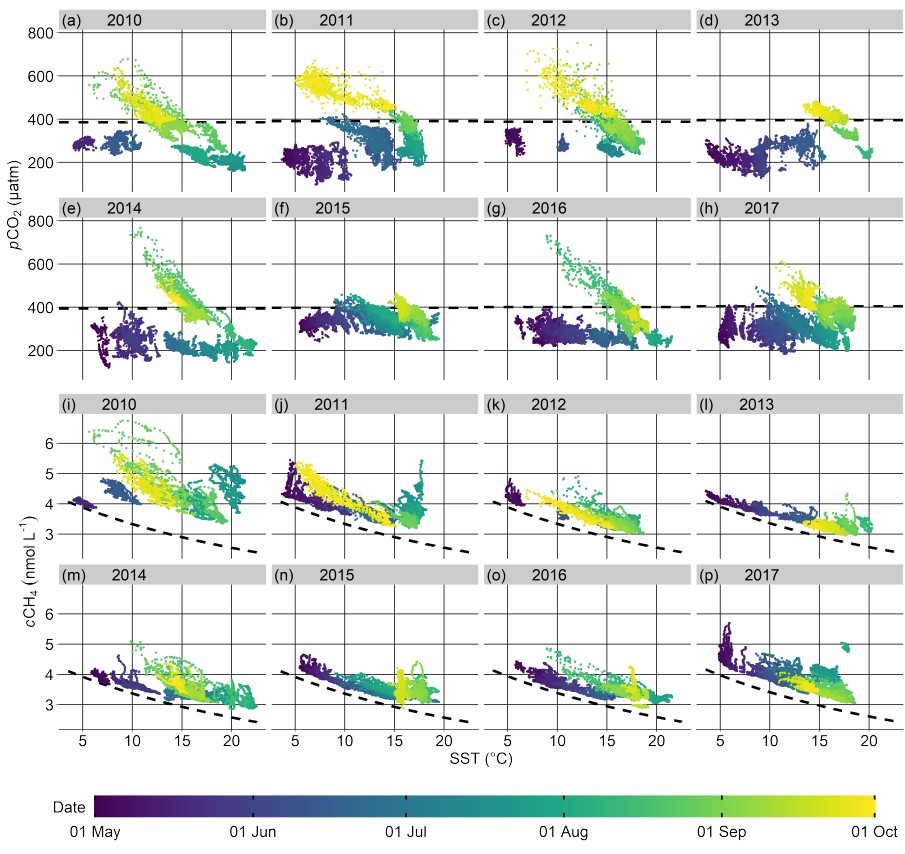

**Figure 10.** Surface $CO_2$ partial pressure (a–h), and $CH_4$ concentration (i–p) from 1 May to 30 September within box Ö-S, each plotted against temperature for individual years. The measurement date is colour-coded. Black dashed lines indicate atmospheric equilibrium partial pressure and concentration (calculated using mean seasonal salinity), respectively.

coloured date scale allows to follow the temporal evolution of signals throughout the season and also highlights larger data
gaps (e.g. in 2012 and 2013).

Most years feature consistent patterns with respect to $pCO_2$ (Fig. 10), reflecting its yearly cycle (see Introduction and Schneider and Müller, 2018): $CO_2$ is already undersaturated with respect to the atmosphere in May due to primary production during the spring bloom. Over the following weeks, the change in $pCO_2$ is usually rather small, but SST increases as a result of solar irradiation and often weaker winds (see also Fig. B4). The resulting stabilisation of the surface thermocline and the
accumulation of remineralised $CO_2$ below combined with decreasing air–sea $CO_2$ exchange and ongoing primary production lead to increasing $pCO_2$ gradients between surface and sub-thermocline water (which are the cause of upwelling-induced $pCO_2$ signals, Fig. 2) and a permanent undersaturation of the surface water with respect to the atmosphere. The characteristic $cCH_4$–SST conditions follow the $CH_4$ saturation curve towards lower concentrations at higher temperatures most probably due to air–sea $CH_4$ exchange, maintaining a persistent supersaturation. In most years, most notably in 2010, 2012, 2014, and 2016,





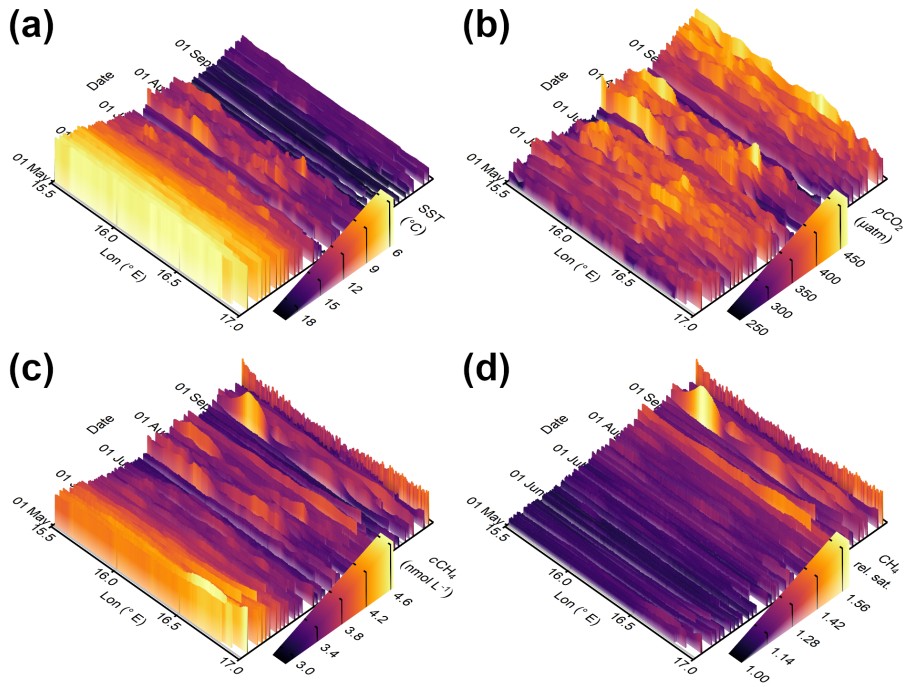

**Figure 11.** (a) Sea surface temperature, (b) $CO_2$ partial pressure, (c) $CH_4$ concentration, and (d) relative $CH_4$ saturation within box Ö-S on 82 transects from 2 May to 21 September 2015. In all panels, abscissa is position given as longitude, ordinate is time, and the respective variable is displayed by both colour and height of the curve. Please note the inverted SST scale in (a) to highlight the correlation between decreasing SST and increasing $c$$CH_4$ and $p$$CO_2$. Data presented in (a–c) correspond to Fig. 10f,n.

strong upwelling around August overrides these typical summer conditions, resulting in characteristic $p$$CO_2$–SST and $c$$CH_4$–SST patterns. For these years, the ranges of SST, $p$$CO_2$, and, to a certain extent, $c$$CH_4$ are similar (but still not equal), with the notable exception of very dynamic $c$$CH_4$ in 2010. For the other years, we observe a high degree of variability from these typical conditions: Strong upwelling-favourable winds in June 2011 led to an early increase of $p$$CO_2$ and lower SST overall. Later, at the end of July 2011, a pronounced, sharp increase of $c$$CH_4$ at rather constant temperature was observed, which clearly is not

related to upwelling.

The year 2015 is particularly interesting because it demonstrates the influence of quasi-continuous upwelling-favourable winds over the course of several months, which overrides the typical summer trace gas situation. The year was dominated by upwelling-favourable, westerly winds until the beginning of August (Fig. A3), effectively prohibiting strong thermal stratification of the surface water (Fig. 11a and low maximum temperature in Fig. 10f,n). This special case is problematic for the

detection method because the observable $\Delta$SST gradients become too small for every day to be counted as an "upwelling day" (Fig. A3). As a result of weakened stratification, surface $CO_2$ undersaturation is unusually weak compared to the typical summer situation (Fig. 11b and high minimum $p$$CO_2$ in Fig. 10f). Furthermore, we observe reduced $c$$CH_4$ variability as a result of continuous mixing and intensified air–sea exchange through increased turbulence, so that $c$$CH_4$ follows the saturation





curve more closely than during most years (Fig. 10n). Elevated $c$CH$_4$ (Fig. 11c) does not necessarily translate into elevated
saturation (Fig. 11d) depending on SST – however, as pointed out in Sect. 3.3, the water mass will become supersaturated as
a consequence of subsequent warming. In July 2015 (turquoise hues in Fig. 10f,n), near-linear trace gas – temperature curves
are characteristic for strong upwelling (see Sect. 3.2). Compared to, e.g. August 2014 and 2016, however, where the upwelling
SST, $p$CO$_2$, and $c$CH$_4$ signals stand out prominently from the rest of the values, their range concerning all three parameters is
reduced in 2015 since decoupling of surface and underlying water was partly impeded.

**3.5    Potential to estimate upwelling-induced air–sea trace gas fluxes**

The observed near-linear relationships between $p$CO$_2$ or $c$CH$_4$ and SST can be used to spatially extrapolate trace gas obser-
vations from sub-transects based on modelled SST fields, assuming that these relationships are consistent for entire upwelling
areas. This enables us to estimate air–sea CO$_2$ and CH$_4$ fluxes resulting from upwelling events (Fig. 12). Since we used linear
regression (Fig. 12f,g), SST minima near the coast translate into $p$CO$_2$ and $c$CH$_4$ maxima, retaining the overall pattern and fine
structure of the SST field (Fig. 12a–c).

The CO$_2$ flux ($F$CO$_2$, Fig. 12d) depends on the difference in $p$CO$_2$ between sea and air, which determines the flux direction,
and the transfer coefficient, which is parametrised mostly by wind speed. Therefore, $p$CO$_2$ and $F$CO$_2$ share the same spatial
pattern with positive and negative fluxes being present in the box at the same time. Both CO$_2$ outgassing (0.13 mol m$^{-2}$ d$^{-1}$)
and uptake (−0.046 mol m$^{-2}$ d$^{-1}$) peak on 9 August, when wind speeds are highest. CH$_4$ fluxes ($F$CH$_4$, Fig. 12e) into the
atmosphere reach their maximum (5730 nmol m$^{-2}$ d$^{-1}$) on the same day. However, the spatial distribution of $F$CH$_4$ differs
from that of $c$CH$_4$ due to the temperature dependence of the CH$_4$ saturation concentration: For instance, the lowest fluxes
on 10 August occur close to the coast despite high concentrations in this area because supersaturation decreases with lower
SST, indicating that the upwelled water mass has a lower CH$_4$ supersaturation than the surrounding waters in this example.
However, relative CH$_4$ saturation is highly sensitive to changes in slope of the applied regression curve, which underestimates
the observed maximum supersaturation in this example (Fig. 12g) and likely leads to this special spatial distribution. Even
careful tweaking of the regression curve would result in a pattern much more similar to that of $F$CO$_2$ and this similarity
increases with increasing supersaturation of the upwelled water mass compared to pre-upwelling conditions. In other examples,
the two flux patterns are more similar than here (data not shown). The discussed "tweaking" of the $c$CH$_4$–SST regression,
though impacting the derived pattern considerably, would have only a small impact on the areal flux. In any case, daily $F$CH$_4$
peak under intermediate conditions where the flux-increasing effects of rising $c$CH$_4$ and rising SST combine. On many days in
this example, including 10 August (Fig. 12e), this area of maximum daily $F$CH$_4$ is in close proximity to the transect of SOOP
*Finnmaid*, where it can be observed by in situ measurements, whereas areas of maximum daily $F$CO$_2$ tend to be closer to the
coast. It should be noted that the spatial variability of $F$CO$_2$ is much higher than that of $F$CH$_4$ (see maximum and minimum
values in Fig. 12i,j).

The importance of wind is also reflected by the evolution of air–sea gas fluxes over time (Fig. 12i,j). Fluxes are weak under
pre-upwelling conditions due to low wind speeds (Fig. 12h), increase distinctly with rising wind speeds, and reach a minimum
(considering absolute values for $F$CO$_2$) in the relaxation period after the upwelling event, when the wind calms down again.



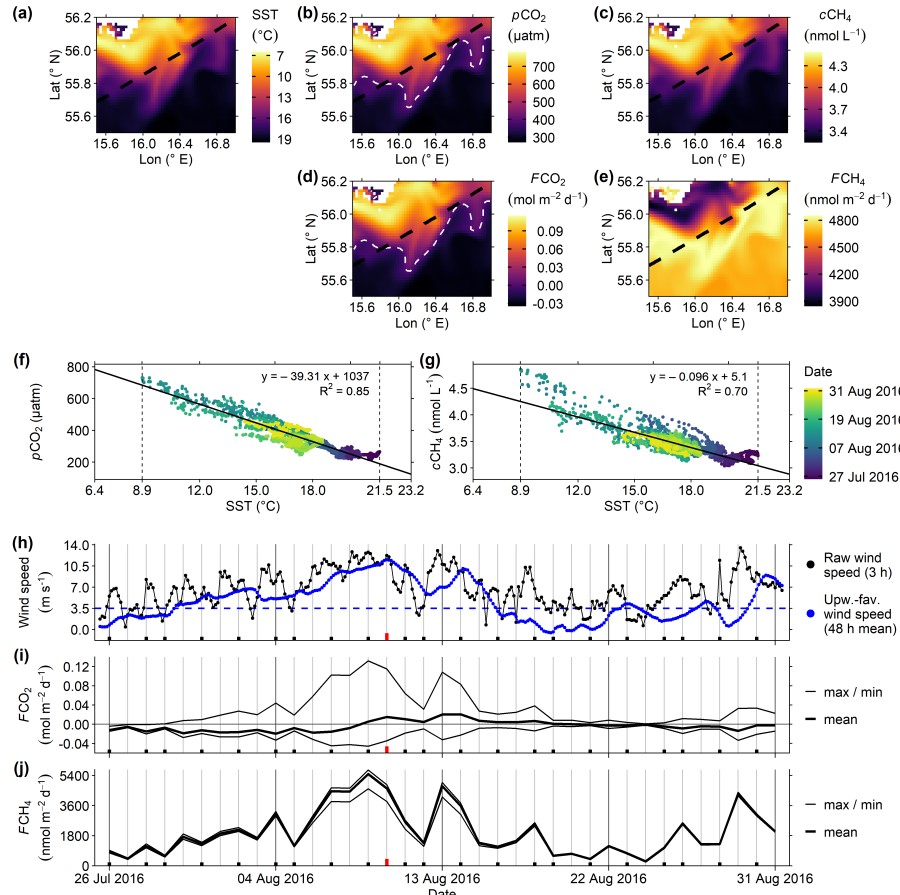

**Figure 12.** Air–sea trace gas flux estimate for region Ö-S from 26 July to 31 August 2016. Maps depict the situation on 10 August, when SST was minimal: (a) Modelled SST (inverted colour scale), based on which (b) $p\mathrm{CO_2}$ and (c) $c\mathrm{CH_4}$ were extrapolated from measurements aboard SOOP *Finnmaid*, leading to air–sea flux estimates of (d) $\mathrm{CO_2}$ and (e) $\mathrm{CH_4}$. Black dashed lines represent the track of SOOP *Finnmaid*, white dashed lines indicate atmospheric equilibrium for $\mathrm{CO_2}$ ($\mathrm{CH_4}$ is always supersaturated). (f) Relationship of $p\mathrm{CO_2}$ or (g) $c\mathrm{CH_4}$ and SST as measured by SOOP *Finnmaid* on 21 sub-transects from 26 July to 30 August 2016 with colour indicating date (same as Fig. 8b,j). Dashed vertical lines display the observed SST range, while the limits of the SST axis represent the range of modelled SST and, thus, the extrapolation limits. The relationships were approximated via linear regression (black solid line), the regression functions and $R^2$ are given in the respective panels. (h) Wind speed (black dots and lines) and the running 48 h mean of upwelling-favourable wind speed (blue dots and lines) over time, with the dashed blue line indicating the threshold of the wind criterion ($3.5\ \mathrm{ms^{-1}}$). (i) Daily air–sea $\mathrm{CO_2}$ and (j) $\mathrm{CH_4}$ fluxes over time. Bold lines represent mean and thin lines max/min fluxes per day, respectively. Black dashes at the bottom of (h–j) mark transects of SOOP Finnmaid within the box; a red, long dash marks 10 August, the date of the maps (a–e).

The sea is a permanent source of atmospheric $\mathrm{CH_4}$ with varying strength based mostly on wind speed in this example, which can be generalised to the entire data set. In contrast, $F\mathrm{CO_2}$ is negative ($\mathrm{CO_2}$ uptake from the atmosphere) under pre-upwelling





445 conditions. $CO_2$ outgassing starts with the onset of upwelling, but at this point, the area is still dominated by increasing $CO_2$ uptake due to rising wind speeds. Both positive and negative $CO_2$ fluxes intensify over the following days, but $CO_2$ outgassing reaches higher absolute values as a result of high $pCO_2$ due to upwelling. The area is a net source of $CO_2$ for the atmosphere (mean $FCO_2 > 0$) from 9 August, when wind speeds are maximal, to 19 August, when $pCO_2$ gradients have sufficiently diminished due to relaxation. Note, however, that mean $FCO_2$ depends on the (arbitrary) choice of box boundaries.

450  The presented flux estimates depend largely on the applicability of the observed near-linear trace gas – SST relationships for the entire area and period, including extrapolation to temperatures lower than the minimum temperature of the trace gas – SST regression (i.e. towards the core of the upwelled water mass, see SST axis in Fig. 12f,g). Verification of this assumption would require a dedicated research cruise involving trace gas measurements perpendicular to the track of SOOP *Finnmaid* with transects towards both coast and open basin. Based on the presented findings, we assume that the extrapolation scheme

455 proposed here underestimates the actual fluxes since the applied regression is based on waters that have already been subject to air–sea gas exchange (see also Sect. 3.3) as opposed to "young" upwelled waters, which are not observed by SOOP *Finnmaid*. Still, the presented method provides means to constrain upwelling-induced trace gas fluxes based on SOOP and model (or, potentially, remote sensing) data on large spatial and temporal scales. We recommend using this approach on a per-event basis to properly calibrate the applied trace gas – SST relationships, which might differ between regions and events.

460 ## 4 Conclusions

Upwelling in the Baltic Sea can be observed using autonomous measurements aboard SOOP, which, compared to dedicated research cruises, provide higher spatial and temporal coverage at the cost of being restricted to surface data and, depending on route, a higher distance to the coast. They enable studies on seasonality, comparison of regions, and observation of processes over long time periods. Combining SOOP-based trace gas measurements with other high-resolution data sets like model or

465 remote sensing data further allows us to a) assess their spatial and temporal representativity by adding information perpendicular to the ferry track, b) assess the prevalence of upwelling even during SOOP data gaps caused by ship schedule and (rare) outages, c) compare events by size, duration, and signal intensity, and d) estimate upwelling-induced air–sea fluxes.

  We found that upwelling is an important process to understand trace gas dynamics in near-coastal environments of the Baltic Sea. Deviations from the usual summer trace gas distribution (as determined for the open basins) are dominated by upwelling

470 in some regions, particularly in coastal areas of the central Baltic Sea, while there appear to be other relevant effects especially towards the Gulf of Finland and around the island of Bornholm. The strongest upwelling-induced trace gas signals in the Baltic Sea occur during intense wind events at the end of summer after a long, relatively calm period of decoupling between surface and underlying water. These strong upwelling events stand out prominently from the otherwise rather uniformly distributed trace gas data in summer and are characterised by near-linear relationships between $pCO_2$ or $cCH_4$ and SST. The relaxation

475 of these upwelling-induced trace gas signals is mainly driven by mixing and modulated by air–sea gas exchange and possibly primary production. Air–sea heat exchange stimulates enhanced supersaturation by warming on a time scale of several weeks.



Interannual variability of upwelling in the Baltic Sea depends on prevailing wind conditions. Half of the years in the data set feature strong upwelling around August, which overrides the typical summer trace gas distributions and leads to values that are unreachable by other means in this season (depending on region for $CH_4$). Quasi-persistent upwelling can prevent strong
stratification and cause untypical values for SST, $pCO_2$, and $cCH_4$, as well as impede the formation of strong vertical gradients.

The observed near-linear relationships between $pCO_2$ or $cCH_4$ and SST suggest extrapolation of trace gas observations based on SST fields from a numerical ocean model (like GETM) or remote sensing. This allows for the estimation of upwelling-induced trace gas fluxes over the course of individual upwelling events, though the validity of this extrapolation of linear trace gas – SST relationships to the core of the upwelling cell requires further verification. The presented results on spatial and
temporal characteristics of upwelling in the Baltic Sea on large scales also enable improved cruise planning to conduct more detailed research on the topic, e.g. extensive research-vessel-based process studies.



## Appendix A: Methods

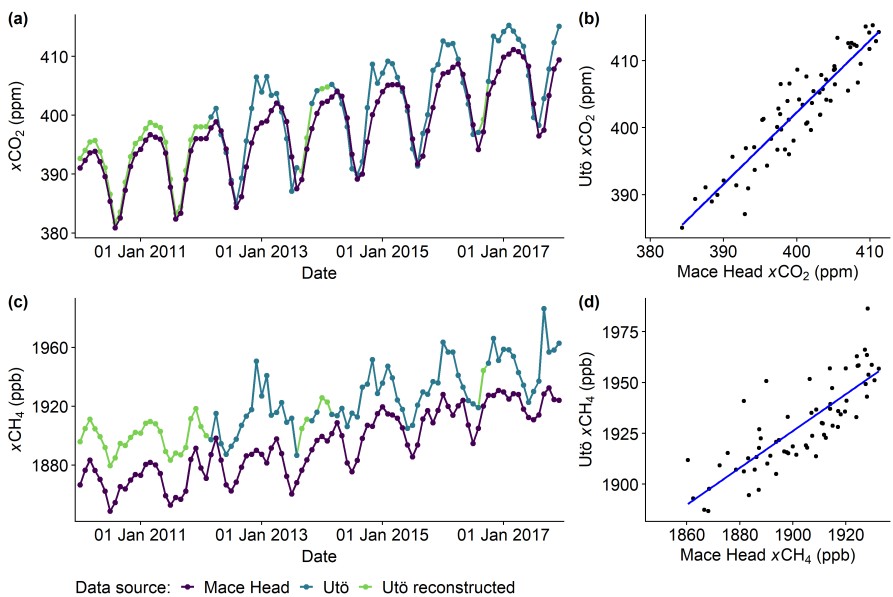

**Figure A1.** Monthly means of atmospheric (a) $CO_2$ and (c) $CH_4$ mole fractions from 2010 to 2017. We preferred data from Utö station (Finnish Meteorological Institute, Helsinki), which start in March 2012, due to their proximity to observations from SOOP *Finnmaid*. Concerning measurements on Utö, the method described in Kilkki et al. (2015) is applicable for the study period except that the Nafion dryer has not been in use since November 2013 (Juha Hatakka, pers. comm.). For the period before 2012 and to fill data gaps in September 2013 and 2016 in the Utö series, we used atmospheric data from Mace Head station (National University of Ireland, Galway) via the NOAA ESRL Carbon Cycle Cooperative Global Air Sampling Network (Dlugokencky et al., 2019a, b), which is roughly at similar latitude. To correct differences between both stations, we normalised the data from Mace Head to those from Utö based on the shared period from 2012 to 2017 using linear regression (b,d). This very simple approach is sufficient to evaluate the magnitude of saturation and compare it on interannual scales.

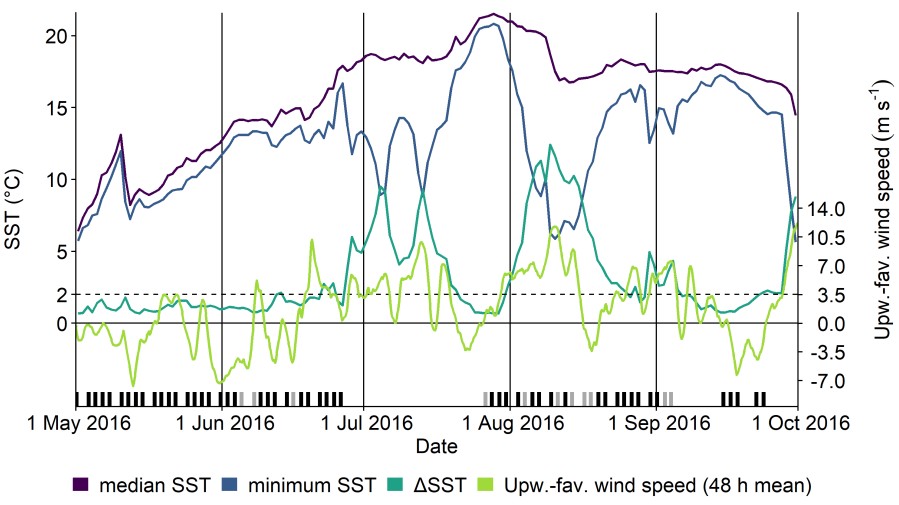

**Figure A2.** As Fig. 4, but based on model-SST data from the entire box instead of along the sub-transect of SOOP *Finnmaid*. Please refer to Sect. 2.3 for a comparison to Fig. 4. Both approaches are further discussed in Sect. 3.1.

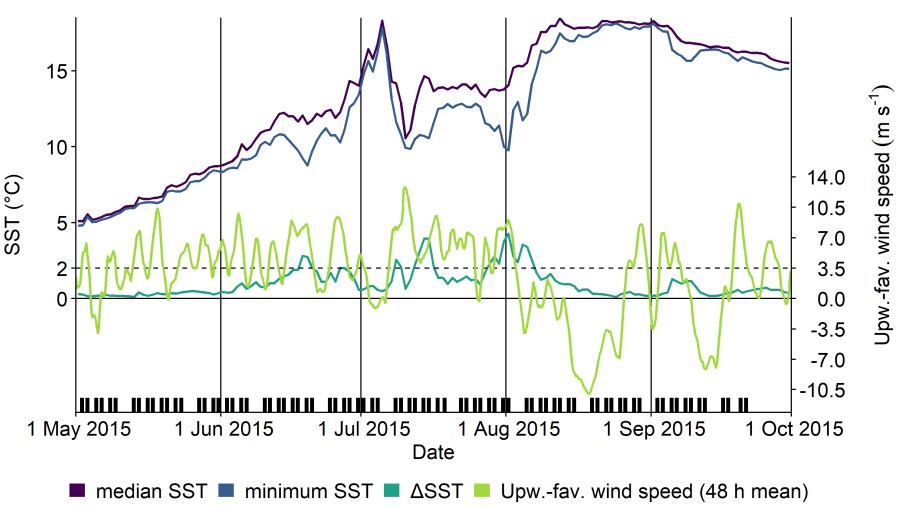

**Figure A3.** As Fig. 4, but within box Ö-S in 2015 (here, both routes go through the box). Quasi-persistent upwelling-favourable wind conditions until the beginning of August effectively prohibited strong thermal stratification of the surface water. This results in lower possible SST gradients from upwelling and, therefore, a rather unreliable ΔSST criterion, which is only triggered during the most intense periods. Please refer to Sect. 3.4 for a detailed discussion of this event.




## Appendix B: Results and discussion

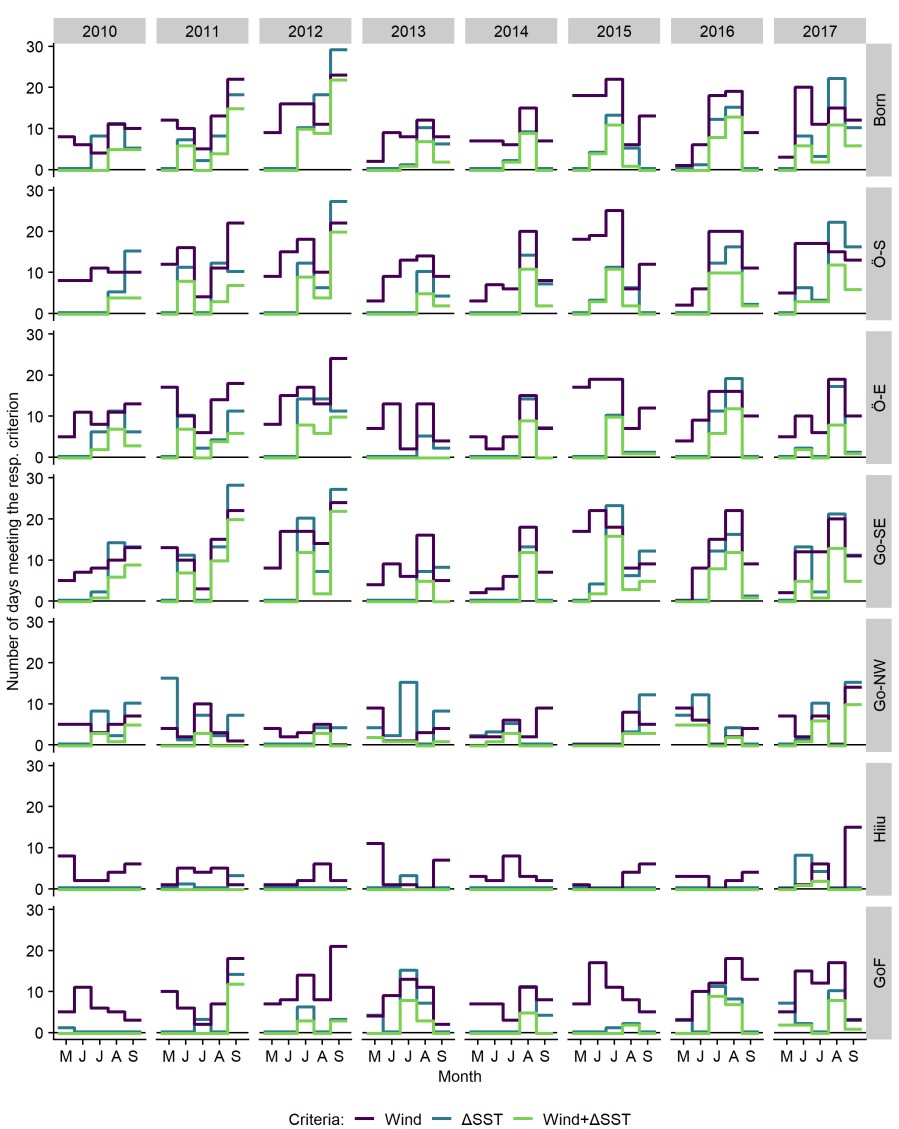

**Figure B1.** As Fig. 6, but further distinguished by year. ΔSST was calculated along the route of SOOP *Finnmaid* through the boxes (Sect. 2.3, second approach). Month abbreviations denote the period May–September.





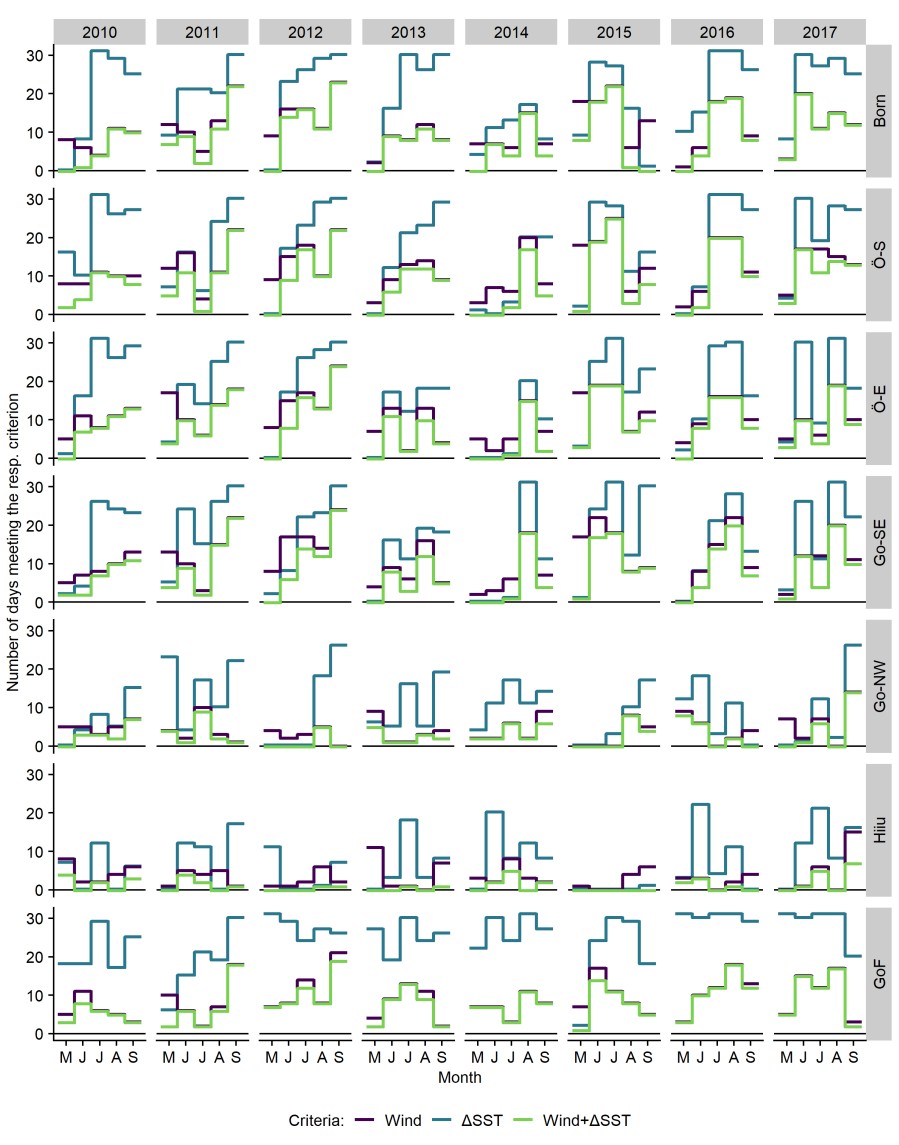

**Figure B2.** As Fig. 6, but further distinguished by year and using a different ΔSST criterion, which was calculated based on the entire area of the boxes (Sect. 2.3, first approach). Month abbreviations denote the period May–September.



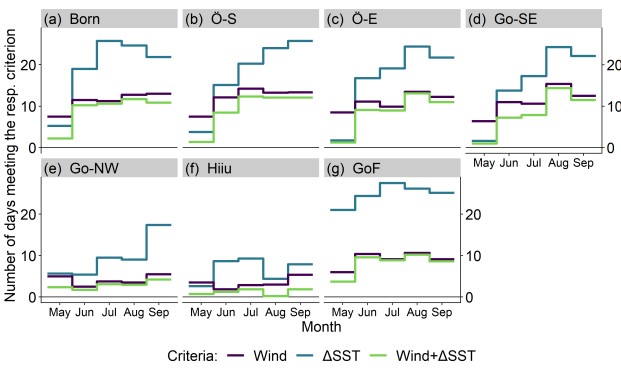

**Figure B3.** As Fig. 6, but using a different ΔSST criterion, which was calculated based on the entire area of the boxes (Sect. 2.3, first approach).



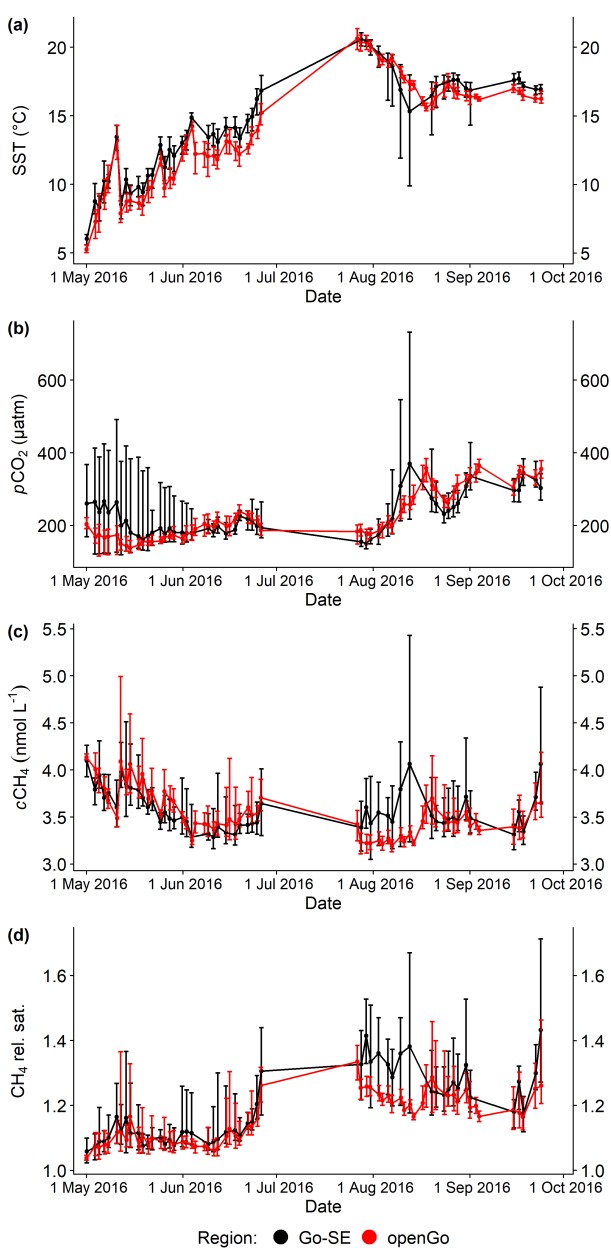

**Figure B4.** (a) Sea surface temperature, (b) $CO_2$ partial pressure, (c) $CH_4$ concentration, and (d) relative $CH_4$ saturation within boxes Go-SE (black) and openGo (red) from 1 May to 23 September 2016. Points and lines represent mean values per sub-transect, error bars denote maximum and minimum, respectively. The effect of upwelling is very prominent in August 2016 (compare Fig. 4), when box Go-SE features decreased SST and increased $pCO_2$, $cCH_4$, and relative $CH_4$ saturation values compared to box openGo, which is not affected by upwelling.



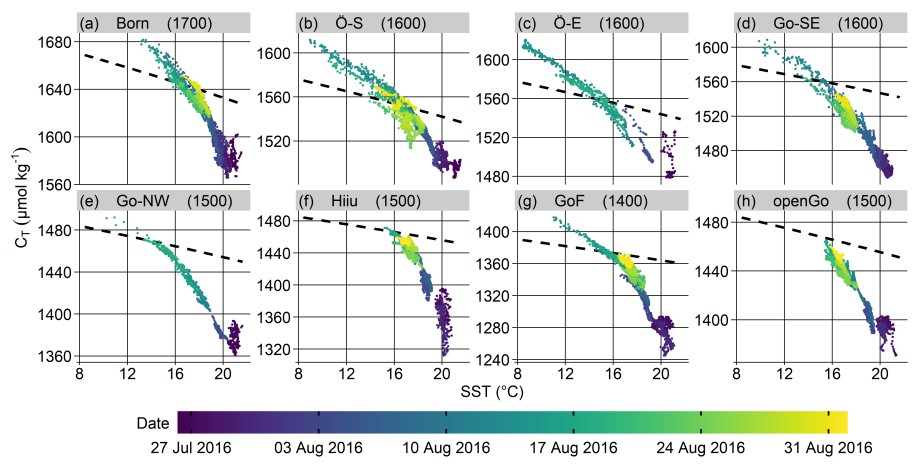

**Figure B5.** Surface $C_T$ estimated from $p$CO$_2$ as measured by SOOP *Finnmaid* on 21 transects from 26 July to 30 August 2016, each plotted against SST within the seven upwelling regions and the open Gotland Sea box for comparison. We estimated $A_T$ for each box (values in $\mu$mol kg$^{-1}$ in parentheses) based on Müller et al. (2016). The measurement date is colour-coded. Temporal coverage in box Go-SE (15 transects) and boxes Ö-E and Go-NW (6 transects) is reduced since SOOP *Finnmaid* uses two different routes around Gotland. Black dashed lines indicate atmospheric equilibrium $C_T$ (calculated using mean salinity per box in the given time period).

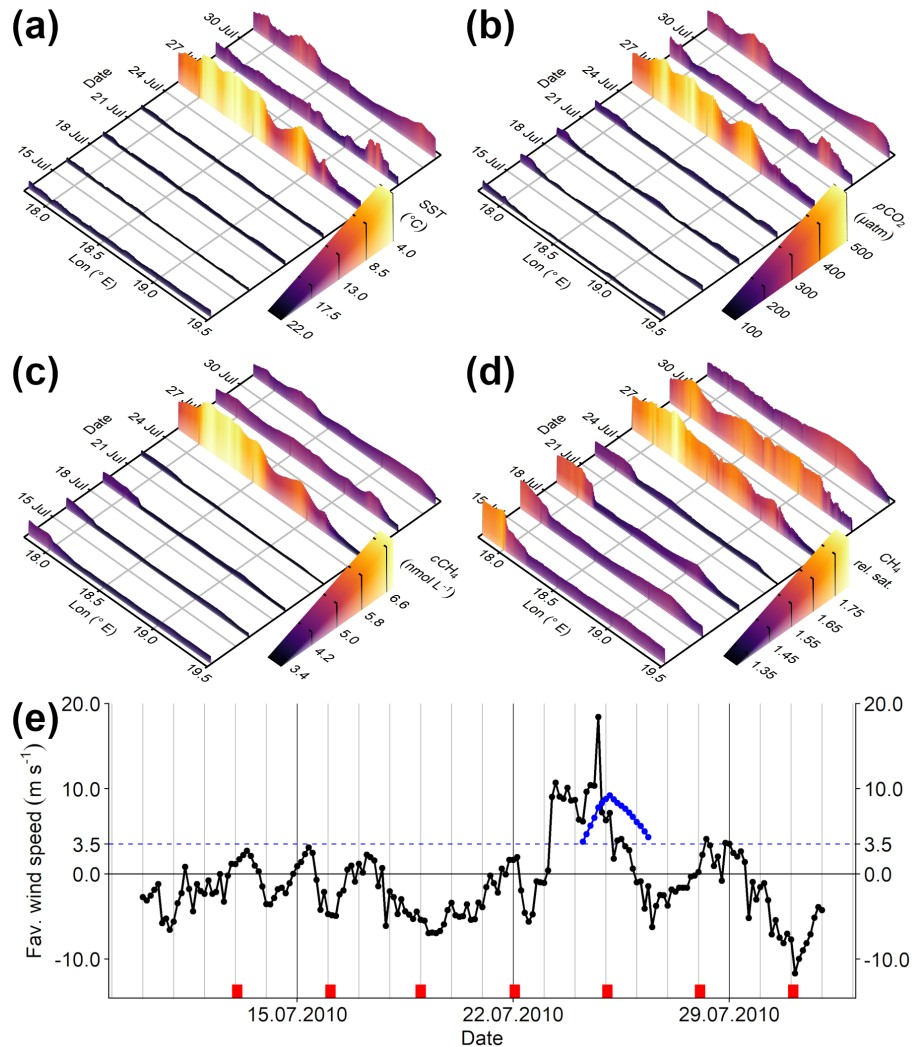

**Figure B6.** (a) Sea surface temperature, (b) $CO_2$ partial pressure, (c) $CH_4$ concentration, and (d) relative $CH_4$ saturation within box Go-NW as measured by SOOP *Finnmaid* on 7 transects from 12 to 31 July 2010. In (a–d), abscissa is position given as longitude, ordinate is time, and the respective variable is displayed by both colour and height of the curve. Please note the inverted SST scale in (a) to highlight the correlation between decreasing SST and increasing $pCO_2$ and $cCH_4$. (e) Upwelling-favourable wind component in the same period: Black dots and lines are in 3 h intervals, blue dots represent the running 48 h mean, if this mean is above 3.5 m s$^{-1}$. The dashed blue line indicates the chosen threshold of the wind criterion. Red dashes at the bottom mark transects of SOOP *Finnmaid* within the box (see (a–d)). This example shows distinct upwelling signals on 25 July, which vanish quickly after turning wind, hinting towards downwelling as the relevant process for fast relaxation of upwelling signals. In fact, post-upwelling values of SST, $pCO_2$, and $cCH_4$ differ only slightly from pre-upwelling values.





*Data availability.* The trace gas data set from SOOP *Finnmaid* will be made available in an openly accessible data base upon final publica-
tion.

*Author contributions.* EJ, GR, and CAG conceived the study. MG, BS, and EJ processed the *Finnmaid* data set and carried out quality control. UG did the model run and provided wind and remote sensing data. HCB, JDM, and CAG provided assistance with analyses. EJ carried out all analyses, prepared the figures, and wrote the manuscript, to which all co-authors provided editorial and scientific recommendations.

*Competing interests.* The authors declare that they have no conflict of interest.

*Acknowledgements.* The authors would like to thank Bernd Sadkowiak for regular maintenance of the measurement setup aboard *Finnmaid*. *Finnmaid* T/S data were provided by Alg@line marine monitoring service at the Finnish Environment Institute (SYKE, Jukka Seppälä). Atmospheric $CO_2$ and $CH_4$ data from Utö station were provided by the Finnish Meteorological Institute (FMI, Juha Hatakka). CAG acknowledges support of her visiting researcher position by the Leibniz Institute for Baltic Sea Research Warnemünde (IOW). This work was funded by the project BONUS INTEGRAL (grant No. 03F0773A), which receives funding from BONUS (Art 185), funded jointly by the
EU, the German Federal Ministry of Education and Research, the Swedish Research Council Formas, the Academy of Finland, the Polish National Centre for Research and Development, and the Estonian Research Council. The measurements were temporarily (2009–2011) funded by the German Federal Ministry of Education and Research in the frame of the BONUS projects Baltic-C (grant No. 03F0486A), Baltic Gas (grant No. 03F0488B), and, since 2012, ICOS-D (grant Nos. 01LK1101F and 01LK1224D).



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
