# Peer review of "Upwelling-induced trace gas dynamics in the Baltic Sea inferred from 8 years of autonomous measurements on a ship of opportunity"

_Biogeosciences, 2020_

## Referee Comment (RC1) · Anonymous Referee #1 · 9 Nov 2020

Carbon dioxide (CO2) and methane (CH4) are climate-relevant trace gases. There-fore, investigations of their distributions as well as estimates of their natural and an-thropogenic sources and sinks have received a lot of attention during the last five decades. In general, the coastal oceans are an overall sink of atmospheric CO2 and an overall source of atmospheric CH4. However, getting a comprehensive picture of the CO2/CH4 distributions in coastal ocean environments is hampered by the fact that the seasonal and interannual variabilities are usually not well known or even unknown. To this end, the manuscript (ms) under review presents underway time series measure-ments of dissolved CO2 and CH4 concentrations from the surface layer of the Baltic Sea made on-board a commercial vessel commuting between Lubeck and Helsinki in

the period from 2010 to 2017. The data set is used to show the effects of coastal upwelling on the distributions and air/sea fluxes of dissolved CO2 and CH4 in various (selected) regions of the Baltic Proper and the Gulf of Finland. Although I think that the ms presents a new data set of high relevance to address questions about seasonality and interannual variability of dissolved CO2 and CH4 in coastal areas such as the Baltic Sea, its major scientific objectives remain unclear. In large parts, the ms reads more like a technical or methodological report and thus needs considerable re-writing. Therefore, I can recommend publication only after significant major revisions.

Major points:

1) The introduction needs significant re-writing. It should give the basic scientific background why this kind of measurements and data analysis are done. Moreover, the overarching scientific objectives addressed by the study need to be given.

2) Section 2 'Data methods': I would like to suggest to move sections 2.2 and 2.4-2.6 to the Appendix. The information given in these sections is relevant only for side aspects of the data analysis. (Please note that Fig. 9 is already mentioned in section 2.4, so the numbering of figures is not correct, it should appear as Fig. 5)

3) Section 3 'Results and Discussion': Coastal upwelling as significant sources of trace gases such as CO2 and CH4 have been found in other coastal systems as well (for example in the eastern boundary upwelling systems off Oregon, Peru, Mauritania, NW Africa). Please discuss the results from the Baltic Sea in light of the results reported in the literature from other coastal upwelling systems. An overview table with saturation/flux data from literature may help to facilitate the comparison.

4) Section 3 'Results and Discussion': I am wondering if the authors could now quantify the significance of the contributions of upwelling-induced CO2/CH4 fluxes to the overall emission estimates of the Baltic Sea. And indeed, on page 18, lines 372-373, I found a statement on this issue saying this '[. . .] still needs further investigation.'. This is rather confusing (and disappointing) since the authors have the data sets at hand to come up

with some numbers to prove the significance.

5) Section 3 'Results and Discussion': Moreover, I am wondering why the authors do not discuss the effects of the ongoing environmental changes of the Baltic Sea (such as warming, changing wind patterns etc.). An important question to be addressed might be: Are there any trends detectable for the upwelling-induced CO2/CH4 fluxes during the course of the study which after all covers eight-years? If yes, what are the main factors causing this trend?

6) Section 4 'Conclusions': It is well-known that CO2 and CH4 are affected by upwelling in the Baltic Sea. This was already shown in publications by the same group (see Gülzow et. al., Biogeosci., 2013; Schneider et al., J Mar Sys., 2014) and thus it surprising to see this stated as a major conclusion (see page 23, 2nd paragraph of section '4 Conclusions').

Minor points:

1) Section 2 'Data and Methods' (and throughout the rest of the text): The authors use the term 'saturation concentration' which is misleading. This term should be replaced with 'equilibrium concentration'.

2) Figure 1: Please indicate the location of the Uto station in the map.

3) P5L101-103: Please note that a concentration is only independent from temperature when it is given as mol kg-1. If it is given as mol L-1 (as in the ms) it is not independent from temperature. Moreover, the partial pressure is depending on the temperature when you refer to the partial pressure in equilibrium with the water phase. Please correct.

---

## Referee Comment (RC2) · Anonymous Referee #2 · 16 Nov 2020

Jacobs et al. present 8-years of underway surface $CO_2$ and $CH_4$ measurements from the Baltic Sea. They assess the role of upwelling on surface gas concentrations and fluxes on seasonal time-scales, and describe typical annual cycles, as well as anomalies. The paper is very well written, and thoroughly describes regional and temporal differences in $CO_2$ and $CH_4$ concentrations, showing clearly the influence of upwelling and temperature. The methods used appear to be robust, and well-explained, with careful consideration of potential sources of error. The dataset itself is of tremendous value, and the interpretation is well-done and could be applied to other regions with underway $CO_2$ and/or $CH_4$ systems.

[Figure]

Although it could be argued the paper lacks clear objectives or motivation, I propose the value of this paper is in the methodological development used to extrapolate discrete underway data, thus improving its already high-resolution. Additionally, the development of a robust technique for identifying upwelling and linking it with observations on these spatiotemporal scales is well done and could be of value to others interpreting similar datasets, which could facilitate more robust extrapolation of such measurements in regions sorely lacking data. This makes for a valuable contribution to understanding the importance of upwelling on temporal and spatial variability in CO2 and CH4 flux, and a delight to read.

I have only minor suggestions to improve clarity of figures (especially regarding choice of colors), and text. I recommend publication of the manuscript.

Fig. 2-. The profile colors are hard to distinguish, and likely would be near impossible for anyone with color-blindness. I suggest using more easily discernable colors.

Fig. 2. Is oxygen available? I suspect it would be relevant especially for CH4.

Line 41: 'two minima' are mentioned, but the subsequent text implies a minima during the spring bloom, and several subsequent minima throughout summer. Not clear when the second surface pCO2 minima typically occurs, or if there are several more? Perhaps revise 'two minima' to '. . .a minimum during spring and one or more subsequent minima throughout summer. . .'

Lines-41-47 – could you clarify when the surface CO2 is typically under-saturated vs. super-saturated when describing the spring/summer surface pCO2?

Line 48-49 – '..vertical redox..' would be helpful to add oxygen profile to fig 2.

Fig. 6 –colors are difficult to distinguish.

Line 332-335 You state that the estimated +0.4degK warming from atmospheric heat flux is insufficient to account for the observed warming. Can you remind the reader what the total surface warming was, and/or how much additional warming needs to be

accounted for by mixing? (this could be included in this paragraph, or in the paragraph at the start of this section that describes temp relaxation after upwelling).

---

## Author Comment (AC1) · 10 Dec 2020

**Thank you very much for the thorough review and your valuable recommendations and suggestions, which surely help to improve the manuscript.**

*Carbon dioxide (CO2) and methane (CH4) are climate-relevant trace gases. Therefore, investigations of their distributions as well as estimates of their natural and anthropogenic sources and sinks have received a lot of attention during the last five decades. In general, the coastal oceans are an overall sink of atmospheric CO2 and an overall source of atmospheric CH4. However, getting a comprehensive picture of the CO2/CH4 distributions in coastal ocean environments is hampered by the fact that the seasonal and interannual variabilities are usually not well known or even unknown. To this end, the manuscript (ms) under review presents underway time series measurements of dissolved CO2 and CH4 concentrations from the surface layer of the Baltic Sea made on-board a commercial vessel commuting between Lubeck and Helsinki in the period from 2010 to 2O17. The data set is used to show the effects of coastal upwelling on the distributions and air/sea fluxes of dissolved CO2 and CH4 in various (selected) regions of the Baltic Proper and the Gulf of Finland. Although I think that the ms presents a new data set of high relevance to address questions about seasonality and interannual variability of dissolved CO2 and CH4 in coastal areas such as the Baltic Sea, its major scientific objectives remain unclear. In large parts, the ms reads more like a technical or methodological report and thus needs considerable re-writing. Therefore, I can recommend publication only after significant major revisions.*

**Thank you for your critical and very helpful assessment. We agree that the scientific objectives of our study have to be stated clearer in the introduction and that some points in the discussion are missing. Please refer to our answers below for detailed comments. In addition to addressing seasonality and interannual variability, we do see the development of techniques for upwelling detection and extrapolation also as important part of the scientific scope of the paper (in agreement with Reviewer #2). Nevertheless, we absolutely agree with you that these objectives have to (and will) be presented more clearly in the introduction.**

*Major points:*

*1) The introduction needs significant re-writing. It should give the basic scientific background why this kind of measurements and data analysis are done. Moreover, the overarching scientific objectives addressed by the study need to be given.*

**There is indeed room for improvement regarding clearly stating what the major objectives of our study are and we agree to rewrite the introduction accordingly. More specifically, we plan to:**

- **Address the climatic and anthropogenic driving forces in the Baltic Sea, which lead to environmental changes (including reference to the BACC II report concerning upwelling frequency), which makes the Baltic Sea a good study site to see feedbacks early and to develop methods that can be used to analyse long-term data sets with respect to e.g. upwelling-induced trace gas dynamics.**
- **Add the issue of bad coverage of seasonality due to (biased) individual RV-based studies in contrast to the full coverage of a SOOP strategy as a motivation of our approach.**
- **Add model character of our study for similar sites and clearly indicating in the discussion which findings we expect to be applicable for the treatise of upwelling in other regions and which are specific for the Baltic Sea.**
- **Sharpening the introduction in general to interconnect all these points, clearly stating what the objectives are and clarifying what is out of scope, also taking your recommendations below into account.**

**We are certain that this rewrite and restructuring of the introduction will improve the manuscript and hope it finds your approval.**

*2) Section 2 'Data methods': I would like to suggest to move sections 2.2 and 2.4-2.6 to the Appendix. The information given in these sections is relevant only for side aspects of the data analysis. (Please note that Fig. 9 is already mentioned in section 2.4, so the numbering of figures is not correct, it should appear as Fig. 5)*

**We follow this suggestion by moving sections 2.4, 2.5, and 2.6 to the appendix, as they address side aspects of the paper or data handling. (This also solves the incorrect figure labelling.)**

**Section 2.2, however, describing wind and model data, is a section dealing with two main input parameters of vital importance to the study. Thus, after thoughtful consideration and discussion, we would like to keep this section in the main part of the manuscript.**

50

*3) Section 3 'Results and Discussion': Coastal upwelling as significant sources of trace gases such as CO2 and CH4 have been found in other coastal systems as well (for example in the eastern boundary upwelling systems off Oregon, Peru, Mauritania, NW Africa). Please discuss the results from the Baltic Sea in light of the results reported in the literature from other coastal upwelling systems. An overview table with saturation/flux data from literature may help to facilitate the comparison.*

55  **Upwelling in the Baltic Sea – compared to oceanic upwelling – is not persistent, but episodic, and admittedly of less importance for the global budget and fluxes. This information is partly in the introduction already, but we intend to expand this, also with respect to the objectives of the study (see above). Our study is focused on $pCO_2$ and $cCH_4$, specifically (i) on their seasonality, interannual variability, and relaxation, (ii) on the drivers and possible feedbacks behind the observed dynamics, and (iii) on providing tools/methods for the community to deal with similar data in**
60  **other upwelling areas, some of which might be more important in a global context. As stated above, we will expand this in the introduction to make our focus clearer. The last section concerning the flux estimate is intended to be an outlook on future perspectives, however, because the resulting fluxes are based on extrapolated rather than measured data and are, thus, in our eyes, not reliable enough to be compared with other data. Therefore, we would like to refrain from a comparison table, which is beyond the scope of the study focusing on the characteristics of**
65  **our data set and methodological advancements.**

*4) Section 3 'Results and Discussion': I am wondering if the authors could now quantify the significance of the contributions of upwelling-induced CO2/CH4 fluxes to the overall emission estimates of the Baltic Sea. And indeed, on page 18, lines 372-373, I found a statement on this issue saying this '[: : :] still needs further investigation.'. This is rather confusing (and disappointing) since*
70  *the authors have the data sets at hand to come up with some numbers to prove the significance.*

**The phrase "needs further investigation" is indeed an unfortunate one. We would like to replace the sentence with: "Despite a more detailed analysis of the statistical prevalence of upwelling in this study, the question of the importance of upwelling on the annual trace gas balance of the Baltic Sea cannot be answered here based on the data available. Apart from the high variability within observed upwelling events, general statements on this matter**
75  **are further complicated by little knowledge about fluxes in shallow areas (Humborg et al., 2019), large heterogeneities between basins (Gülzow et al., 2013), and the unknown $CO_2$ source/sink behaviour of the entire Baltic Sea (Schneider et al., 2014b). Answering this question in the future requires more knowledge on the Baltic Sea $CO_2/CH_4$ balances in general and extended insight into limitations of upwelling-induced flux estimates in the Baltic Sea (discussed in Sect. 3.5)." This should give proper justification to our statement. As stated above, we will**
80  **also make clearer in the introduction that our main objectives are to find controls of the observed variability and dynamics within eight years, and to show ways to deal with extremely variable conditions in long-term data sets.**

*5) Section 3 'Results and Discussion': Moreover, I am wondering why the authors do not discuss the effects of the ongoing environmental changes of the Baltic Sea (such as warming, changing wind patterns etc.). An important question to be addressed*
85  *might be: Are there any trends detectable for the upwelling-induced CO2/CH4 fluxes during the course of the study which after all covers eight-years? If yes, what are the main factors causing this trend?*

**Indeed, looking for trends in the data set was one objective of ours, which we will stress in the introduction, including information on environmental changes in the Baltic Sea. However, we did not find any trends on time scales of a few years as a result of the large variabilities that surpass any trends that might exist. This information**
90  **is missing in the manuscript so far, so we added a short paragraph on this matter at the end of section 3.4: "It was not possible to identify trends in frequency or magnitude of enhanced $pCO_2$ and $cCH_4$ caused by upwelling events on time scales of a few years. The main reason for this are the high spatial and temporal variability of upwelling (and of several other processes with influence on dissolved trace gases) in the Baltic Sea, which surpass any trend**

that might exist. Moreover, the observed endmembers of minimum SST and maximum $p\mathrm{CO_2}/c\mathrm{CH_4}$ are dependent on data coverage, which adds another layer of uncertainty to any trend analysis on the data set, especially in boxes around Gotland (two different ship routes) and during years with larger data gaps." We further plan to address typical/dominant time scales in the Baltic Sea, the well-above-global-mean warming of the Baltic Sea, and both potential mechanisms for enhanced $\mathrm{CH_4}$ and $\mathrm{CO_2}$ production as well as potential changes in upwelling intensity will be discussed.

*6) Section 4 'Conclusions': It is well-known that CO2 and CH4 are affected by upwelling in the Baltic Sea. This was already shown in publications by the same group (see Gülzow et. al., Biogeosci., 2013; Schneider et al., J Mar Sys., 2014) and thus it surprising to see this stated as a major conclusion (see page 23, 2nd paragraph of section '4 Conclusions').*

**The criticised sentence was intended to provide context for the following paragraph, but you are right that it is an unfortunate choice of words. We will replace it with: "Based on the long-term SOOP data set, we identified controlling parameters of upwelling-induced trace gas dynamics in the Baltic Sea on large spatial and temporal scales:" This should indicate clearer which advancements have been made compared to previous studies.**

*Minor points:*

*1) Section 2 'Data and Methods' (and throughout the rest of the text): The authors use the term 'saturation concentration' which is misleading. This term should be replaced with 'equilibrium concentration'.*

**Equilibrium concentration is in fact the more suitable expression, thank you. We will replace it accordingly throughout the manuscript, but will stick to the phrases "supersaturated / undersaturated" as these are clear within the context (we defined relative saturation as ratio of $c\mathrm{CH_4}$ to equilibrium concentration) and are used frequently in the literature in similar context.**

*2) Figure 1: Please indicate the location of the Uto station in the map.*

**Done.**

*3) P5L101-103: Please note that a concentration is only independent from temperature when it is given as mol kg-1. If it is given as mol L-1 (as in the ms) it is not independent from temperature. Moreover, the partial pressure is depending on the temperature when you refer to the partial pressure in equilibrium with the water phase. Please correct.*

**We will correct/clarify both statements. We plan to replace with: "Note that – neglecting the very small influence of temperature on water density – we can handle the concentration (of $\mathrm{CH_4}$ in nmol $\mathrm{L^{-1}}$) as a quasi-conservative parameter with respect to temperature changes in the discussion. In contrast, the partial pressure (of $\mathrm{CO_2}$) in equilibrium with the water phase is temperature-dependent (see also Sect. 3.3)."**

---

## Author Comment (AC2) · 10 Dec 2020

**Thank you very much for the thorough review and your valuable recommendations and suggestions, which surely help to improve the manuscript.**

*Jacobs et al. present 8-years of underway surface CO2 and CH4 measurements from the Baltic Sea. They assess the role of upwelling on surface gas concentrations and fluxes on seasonal time-scales, and describe typical annual cycles, as well as anomalies. The paper is very well written, and thoroughly describes regional and temporal differences in CO2 and CH4 concentrations, showing clearly the influence of upwelling and temperature. The methods used appear to be robust, and well-explained, with careful consideration of potential sources of error. The data set itself is of tremendous value, and the interpretation is well-done and could be applied to other regions with underway CO2 and/or CH4 systems.*

*Although it could be argued the paper lacks clear objectives or motivation, I propose the value of this paper is in the methodological development used to extrapolate discrete underway data, thus improving its already high-resolution. Additionally, the development of a robust technique for identifying upwelling and linking it with observations on these spatiotemporal scales is well done and could be of value to others interpreting similar data sets, which could facilitate more robust extrapolation of such measurements in regions sorely lacking data. This makes for a valuable contribution to understanding the importance of upwelling on temporal and spatial variability in CO2 and CH4 flux, and a delight to read.*

**Thank you for the encouraging words. The objectives you mentioned are indeed main foci of our study. As Reviewer #1 pointed out, however, we will point out our intended main goals of the study clearer in the introduction.**

*I have only minor suggestions to improve clarity of figures (especially regarding choice of colors), and text. I recommend publication of the manuscript.*

*Fig. 2-. The profile colors are hard to distinguish, and likely would be near impossible for anyone with color-blindness. I suggest using more easily discernable colors.*

**We have spent many thoughts on this topic and all colour scales in the manuscript have been chosen with the colour-blindness issue in mind. The colour scales we used ([https://cran.r-project.org/web/packages/viridis/vignettes/intro-to-viridis.html](https://cran.r-project.org/web/packages/viridis/vignettes/intro-to-viridis.html)) are perceptionally uniform and robust to both colour-blindness *and* grey-scale printing. We would like to keep all those advantages. However, we do agree that the differences between colours are rather small in Fig. 2, which is partly because we left out yellow. You will find another version of the plot below, which includes yellow (thereby widens the colour gradient) and has bolder lines to make the colours more distinguishable. We hope that this improved version will find your approval.**

[Figure]

*Fig. 2. Is oxygen available? I suspect it would be relevant especially for CH4.*

**Figure 2 already contains an oxygen profile. However, we do realize that we did not define "$cO_2$" because we do not need the term elsewhere and that it is rather easy to be confused with "$CO_2$". Therefore, we replaced the legend label with "$O_2$ concentration" to make it clearer (see above).**

*Line 41: 'two minima' are mentioned, but the subsequent text implies a minima during the spring bloom, and several subsequent minima throughout summer. Not clear when the second surface pCO2 minima typically occurs, or if there are several more? Perhaps revise 'two minima' to ': : :a minimum during spring and one or more subsequent minima throughout summer: : :'*

**Indeed, the two sentence parts do not really fit together and we revised the sentence according to your recommendation: "The typical seasonality of surface $CO_2$ partial pressure ($p$CO$_2$) in the Baltic Sea features a minimum during spring and one or more subsequent minima throughout summer: The spring bloom starts in March/April…"**

*Lines-41-47 – could you clarify when the surface CO2 is typically under-saturated vs. super-saturated when describing the spring/summer surface pCO2?*

**We added "and undersaturation is usually observed from March/April to September/October, depending on region (Schneider and Müller, 2018)." to the end of the paragraph.**

*Line 48-49 – '..vertical redox..' would be helpful to add oxygen profile to fig 2.*

**Please see above comment addressing Figure 2.**

*Fig. 6 –colors are difficult to distinguish.*

**As outlined above, the colour scales we used follow all criteria and many thoughts have been spent on them. We would like to keep it as is, since there are only three colours in this example. We will make the legend labels bolder, though, as these are indeed a little hard to distinguish.**

*Line 332-335 You state that the estimated +0.4degK warming from atmospheric heat flux is insufficient to account for the observed warming. Can you remind the reader what the total surface warming was, and/or how much additional warming needs to be accounted for by mixing? (this could be included in this paragraph, or in the paragraph at the start of this section that describes temp relaxation after upwelling).*

**We will clarify this and replace "the observed warming of upwelled water masses" with "the observed warming of upwelled water masses in the order of 5–10 K".**

---

## Author Response (AR1)

**Author's Response to Reviewer #1**

**Thank you very much for the thorough review and your valuable recommendations and suggestions, which surely helped to improve the manuscript.**

*Carbon dioxide (CO2) and methane (CH4) are climate-relevant trace gases. Therefore, investigations of their distributions as well as estimates of their natural and anthropogenic sources and sinks have received a lot of attention during the last five decades. In general, the coastal oceans are an overall sink of atmospheric CO2 and an overall source of atmospheric CH4. However, getting a comprehensive picture of the CO2/CH4 distributions in coastal ocean environments is hampered by the fact that the seasonal and*

10 *interannual variabilities are usually not well known or even unknown. To this end, the manuscript (ms) under review presents underway time series measurements of dissolved CO2 and CH4 concentrations from the surface layer of the Baltic Sea made on-board a commercial vessel commuting between Lubeck and Helsinki in the period from 2010 to 2O17. The data set is used to show the effects of coastal upwelling on the distributions and air/sea fluxes of dissolved CO2 and CH4 in various (selected) regions of the Baltic Proper and the Gulf of Finland. Although I think that the ms presents a new data set of high relevance to address questions*

15 *about seasonality and interannual variability of dissolved CO2 and CH4 in coastal areas such as the Baltic Sea, its major scientific objectives remain unclear. In large parts, the ms reads more like a technical or methodological report and thus needs considerable re-writing. Therefore, I can recommend publication only after significant major revisions.*

**Thank you for your critical and very helpful assessment. We agree that the scientific objectives of our study had to be stated clearer in the introduction and that some points in the discussion were missing. Please refer to our**

20 **answers below for detailed comments. In addition to addressing seasonality and interannual variability, we do see the development of techniques for upwelling detection and extrapolation also as important part of the scientific scope of the paper (in agreement with Reviewer #2). Nevertheless, we absolutely agree with you that these objectives have to be (and are now) presented more clearly in the introduction.**

25 *Major points:*

*1) The introduction needs significant re-writing. It should give the basic scientific background why this kind of measurements and data analysis are done. Moreover, the overarching scientific objectives addressed by the study need to be given.*

**There was indeed room for improvement regarding the scientific background of the analyses and clearly stating what the major objectives of our study are. We rewrote and restructured the introduction accordingly.**

30 **We started with an introduction on coastal upwelling (not only in the Baltic Sea) and on how it relates to greenhouse gases, mentioning important processes and ongoing changes as a motivation to measure greenhouse gases:**

**"Coastal upwelling areas are known to be hotspots of greenhouse gas emissions from marine systems to the atmosphere (Capelle and Tortell, 2016; Morgan et al., 2019). This is a result of enhanced productivity and mineralisation, often hypoxic conditions in the subsurface waters leading to increased concentrations of reduced**

35 **compounds, and an effective transport mechanism of these subsurface waters to the air–sea interface. Climate change is expected to have an amplifying effect on the intensity of coastal upwelling in the global ocean (Bakun et al., 2015; Xiu et al., 2018). Eutrophication and reduced ventilation lead to the spreading and intensification of hypoxia in coastal systems (Diaz and Rosenberg, 2008). The combination of these effects will undoubtedly affect the trace gas dynamics in upwelling regions." (Line 21)**

40 **We continued with ongoing environmental changes in the Baltic Sea (which you also mentioned in point 5). This highlights not only the importance of our measurements "out of local interest", but gives good reasons to see the Baltic Sea as a "test lab" for other regions because changes can probably be spotted early here and methods can be developed accordingly:**

**"The Baltic Sea – a brackish, semi-enclosed sea in northern Europe (Fig. 1) – is among the fastest-warming marginal**

45 **seas world-wide (Kniebusch et al., 2019). It is further known to be strongly affected by environmental changes such**

as eutrophication (HELCOM, 2018), changing wind patterns, and, thus, upwelling intensity (BACC II Author Team, 2015), and has also been shown to encounter a decrease of oxygen on a large number of coastal sites over the last decades (Caballero-Alfonso et al., 2015). The Baltic Sea, therefore, offers the potential to study feedbacks between anthropogenic and climatic drivers and upwelling-induced greenhouse gas fluxes early and with a high signal intensity." **(Line 28)**

The following paragraphs were only slightly changed, based on, e.g. suggestions by Reviewer #2. We moved the paragraph on SOOPs and on the instrumentation (which was the first one before) towards the end of the introduction.

We then summarise our motivation and give reasons to do these kind of studies and develop new methods in the Baltic Sea, using SOOP data:

"The large extent to which the Baltic Sea is influenced by climatic and anthropogenic forces and the availability of the presented eight-years data set of SOOP Finnmaid and of high-resolution models (Placke et al., 2018; Gräwe et al., 2019) make the Baltic Sea a unique study site to detect feedbacks early and to develop methods and process understanding that can be used to analyse long-term data sets with respect to, e.g. upwelling-induced trace gas dynamics. The SOOP strategy allows us to investigate the influence of coastal upwelling on surface $p$CO$_2$ and $c$CH$_4$ in the Baltic Sea on a large spatial and temporal scale without issues of bad coverage of seasonality due to (biased) individual RV-based studies. Furthermore, methods developed here can possibly be used for the treatise of upwelling in regions that are more important for global trace gas fluxes and budgets." **(Line 90)**

As requested, we revised the presentation of our objectives, as well:

In this study, we:

- present a method to identify upwelling events along the SOOP track based on wind and modelled SST data,
- compare upwelling-induced trace gas dynamics within several regions in the Baltic Sea,
- examine the relaxation of upwelling events over time with a focus on underlying processes,
- discuss interannual variability within the data set with a focus on controlling mechanisms of seasonality and highlight the importance of upwelling to understand CO$_2$ and CH$_4$ dynamics in the Baltic Sea,
- test whether a long-term trend can be inferred from the analysis of this eight-years data set, and
- demonstrate the potential of extrapolating trace gas observations based on modelled SST data and show ways to estimate air–sea trace gas fluxes on a broader spatial scale under extremely variable conditions.

**(Line 97)**

*2) Section 2 'Data methods': I would like to suggest to move sections 2.2 and 2.4-2.6 to the Appendix. The information given in these sections is relevant only for side aspects of the data analysis. (Please note that Fig. 9 is already mentioned in section 2.4, so the numbering of figures is not correct, it should appear as Fig. 5)*

We followed this suggestion by moving sections 2.4, 2.5, and 2.6 to the appendix (now sections A1, A2, B1), as they address side aspects of the paper or data handling. (This also solves the incorrect figure labelling.)

Section 2.2, however, describing wind and model data, is a section dealing with two main input parameters of vital importance to the study. Thus, after thoughtful consideration and discussion, we would like to keep this section in the main part of the manuscript.

*3) Section 3 'Results and Discussion': Coastal upwelling as significant sources of trace gases such as CO2 and CH4 have been found in other coastal systems as well (for example in the eastern boundary upwelling systems off Oregon, Peru, Mauritania, NW Africa). Please discuss the results from the Baltic Sea in light of the results reported in the literature from other coastal upwelling systems. An overview table with saturation/flux data from literature may help to facilitate the comparison.*

Upwelling in the Baltic Sea – compared to oceanic upwelling – is not persistent, but episodic, and admittedly of less importance for the global budget and fluxes. We expanded this information in the introduction, also with respect to the objectives of the study (see above). Our study is focused on $pCO_2$ and $cCH_4$, specifically (i) on their seasonality, interannual variability, and relaxation, (ii) on the drivers and possible feedbacks behind the observed dynamics, and (iii) on providing tools/methods for the community to deal with similar data in other upwelling areas, some of which might be more important in a global context. The last section concerning the flux estimate is intended to be an outlook on future perspectives, however, because the resulting fluxes are based on extrapolated rather than measured data and are, thus, in our eyes, not reliable enough to be compared with other data. Therefore, we would like to refrain from a comparison table, which is beyond the scope of the study focusing on the characteristics of our data set and methodological advancements.

To address this – apart from improvements in the introduction – we also clarified our focus on the methods and their potential applicability at the end of Sect. 3.5:

"the presented method provides means to constrain upwelling-induced trace gas fluxes based on SOOP and model (or, potentially, remote sensing) data on large spatial and temporal scales, which may also be used to study other upwelling areas with potentially greater impact on global trace gas fluxes than the Baltic Sea. […] In general, this extrapolation method should be applicable in every upwelling area world-wide where near-linear trace gas – SST relationships are observed and could, therefore, be a valuable tool to produce flux maps from (scarce) surface observations." (Line 455)

We also added the following sentence to the conclusion to emphasise the need to prove the method's validity before comparing these extrapolated data with other published fluxes:

"If future investigations show that freshly upwelled waters near the coast possess similar characteristics as those observed from the SOOP, a then well-founded flux estimate will properly constrain upwelling-induced $CO_2$ and $CH_4$ fluxes in the Baltic Sea and enable to relate the trace gas flux magnitude caused by upwelling to total annual flux estimates, as well as a robust comparison with other upwelling regions on a global scale." (Line 491)

We further stressed the potential applicability of our methods to other upwelling areas at the end of the conclusion:

"Furthermore, the detection and extrapolation methods presented here could be applicable in other upwelling areas, which are more relevant on a global scale regarding trace gas fluxes and balances, but lack an appropriate data coverage." (Line 498)

With these changes, we are certain that – although we were not able to do a flux comparison with a total annual flux – we clearly demonstrated the way towards this goal and that we gave good reasons for this choice. With the presented methods, we think that our manuscript is worth reading for the international community nonetheless.

*4) Section 3 'Results and Discussion': I am wondering if the authors could now quantify the significance of the contributions of upwelling-induced CO2/CH4 fluxes to the overall emission estimates of the Baltic Sea. And indeed, on page 18, lines 372-373, I found a statement on this issue saying this '[: : :] still needs further investigation.'. This is rather confusing (and disappointing) since the authors have the data sets at hand to come up with some numbers to prove the significance.*

The phrase "needs further investigation" was indeed an unfortunate one. We replaced the sentence with:

"Despite a more detailed analysis of the statistical prevalence of upwelling in this study, the question of the importance of upwelling on the annual trace gas balance of the Baltic Sea cannot be answered here based on the data available. Apart from the high variability within observed upwelling events, general statements on this matter are further complicated by little knowledge about fluxes in shallow areas (Humborg et al., 2019), large heterogeneities between basins (Gülzow et al., 2013), and the unknown $CO_2$ source/sink behaviour of the entire Baltic Sea (Schneider et al., 2014b). Answering this question in the future requires more knowledge on the Baltic Sea $CO_2/CH_4$ balances in general and extended insight into limitations of upwelling-induced flux estimates in the Baltic Sea (discussed in Sect. 3.5)." (Line 352)

**This should give proper justification to our statement. As stated above, we also made clearer in the introduction that our main objectives are to find controls of the observed variability and dynamics within eight years, and to show ways to deal with extremely variable conditions in long-term data sets.**

*5) Section 3 'Results and Discussion': Moreover, I am wondering why the authors do not discuss the effects of the ongoing environmental changes of the Baltic Sea (such as warming, changing wind patterns etc.). An important question to be addressed might be: Are there any trends detectable for the upwelling-induced CO2/CH4 fluxes during the course of the study which after all covers eight-years? If yes, what are the main factors causing this trend?*

**Indeed, looking for trends in the data set was one objective of ours, which we now added in the introduction, including information on environmental changes in the Baltic Sea (see above). However, this analysis was difficult due to several circumstances, which we now explain at the end of Sect. 3.4:**

**"It was not possible to identify trends in frequency or magnitude of enhanced $p$CO$_2$ and $c$CH$_4$ caused by upwelling events on the limited time scale of eight years covered by our observations. The main reason for this is the high spatial and temporal variability of upwelling (and of several other processes with influence on dissolved trace gases) in the Baltic Sea, which led to the necessity to do parts of the analyses on a per-event basis and effectively impeded a universal approach. Moreover, the observed endmembers of minimum SST and maximum $p$CO$_2$/$c$CH$_4$ are dependent on data coverage, which adds another layer of uncertainty to any trend analysis on the data set, especially in boxes around Gotland (two different ship routes) and during years with larger data gaps. Typical water residence times of 10–30 years (Feistel et al., 2010) imply that longer trace gas time series are needed to detect trends other than variability using the methods we presented here. In fact, Schneider and Müller (2018) managed to find a trend in surface $p$CO$_2$ between 4.6 and 6.1 μatm year$^{-1}$ in the Baltic Sea from 2008 to 2015, but did so without a focus on upwelling events only and by filling data gaps through interpolation, which effectively yielded a considerably larger data basis compared to this study. We suppose that if, at some point, the extrapolation scheme proposed in Sect. 3.5 could be expanded to cover more than single events, a trend analysis based on the resulting $p$CO$_2$/$c$CH$_4$ fields should be possible, since these data would not be restricted by spatial-temporal coverage or event-specific features." (Line 400)**

**We also added the following to the conclusion:**

**"The observed high variability combined with uncertainties from data coverage prevented a detailed trend analysis since parts of the study are still limited to single events. Further, long water residence times (10–30 years) characteristic for the Baltic Sea require equally large data sets. Here, the presented detection and extrapolation methods might facilitate trend analysis in the future by providing $p$CO$_2$/$c$CH$_4$ fields that are not limited by spatial-temporal coverage or event-specific features." (Line 484)**

**We added the following to the abstract:**

**"Trend analysis is still prevented by the observed high variability, uncertainties from data coverage, and long water residence times of 10–30 years." (Line 14)**

**We also added more context concerning environmental changes in the Baltic Sea to the introduction, as explained above.**

*6) Section 4 'Conclusions': It is well-known that CO2 and CH4 are affected by upwelling in the Baltic Sea. This was already shown in publications by the same group (see Gülzow et. al., Biogeosci., 2013; Schneider et al., J Mar Sys., 2014) and thus it surprising to see this stated as a major conclusion (see page 23, 2nd paragraph of section '4 Conclusions').*

**The criticised sentence was intended to provide context for the following paragraph, but you are right that it was an unfortunate choice of words. We replaced it with:**

**"Based on the long-term SOOP data set, we identified controlling parameters of upwelling-induced trace gas dynamics in the Baltic Sea on large spatial and temporal scales:" (Line 470)**

This should indicate clearer which advancements have been made compared to previous studies. We also removed the corresponding statement in the abstract.

*Minor points:*

*1) Section 2 'Data and Methods' (and throughout the rest of the text): The authors use the term 'saturation concentration' which is misleading. This term should be replaced with 'equilibrium concentration'.*

Equilibrium concentration is in fact the more suitable expression, thank you. We replaced it accordingly throughout the manuscript, but stuck to the phrases "supersaturated / undersaturated" as these are clear within the context (we defined relative saturation as ratio of $c$CH$_4$ to equilibrium concentration) and are used frequently in the literature in similar context.

*2) Figure 1: Please indicate the location of the Uto station in the map.*

Done.

*3) P5L101-103: Please note that a concentration is only independent from temperature when it is given as mol kg-1. If it is given as mol L-1 (as in the ms) it is not independent from temperature. Moreover, the partial pressure is depending on the temperature when you refer to the partial pressure in equilibrium with the water phase. Please correct.*

We corrected/clarified both statements. We replaced with:

"Note that – neglecting the very small influence of temperature on water density – we can handle the concentration (of CH$_4$ in nmol L$^{-1}$) as a quasi-conservative parameter with respect to temperature changes in the discussion. In contrast, the partial pressure (of CO$_2$) in equilibrium with the water phase is temperature-dependent (see also Sect. 3.3)." (Line 122)

At the end, we have the feeling that the manuscript has improved a lot as a result of your suggestions. The rewritten introduction should give better context and orientation than the old one, and the added information to discussion and conclusion should give more insight into the details and perspectives of our work. Thank you again for your valuable and helpful review.

**Author's Response to Reviewer #2**

Thank you very much for the thorough review and your valuable recommendations and suggestions, which surely helped to improve the manuscript.

*Jacobs et al. present 8-years of underway surface CO2 and CH4 measurements from the Baltic Sea. They assess the role of upwelling on surface gas concentrations and fluxes on seasonal time-scales, and describe typical annual cycles, as well as anomalies. The paper is very well written, and thoroughly describes regional and temporal differences in CO2 and CH4 concentrations, showing clearly the influence of upwelling and temperature. The methods used appear to be robust, and well-explained, with careful consideration of potential sources of error. The data set itself is of tremendous value, and the interpretation is well-done and could be applied to other regions with underway CO2 and/or CH4 systems.*

*Although it could be argued the paper lacks clear objectives or motivation, I propose the value of this paper is in the methodological development used to extrapolate discrete underway data, thus improving its already high-resolution. Additionally, the development of a robust technique for identifying upwelling and linking it with observations on these spatiotemporal scales is well done and could be of value to others interpreting similar data sets, which could facilitate more robust extrapolation of such measurements in regions sorely lacking data. This makes for a valuable contribution to understanding the importance of upwelling on temporal and spatial variability in CO2 and CH4 flux, and a delight to read.*

Thank you for the encouraging words. The objectives you mentioned are indeed main foci of our study. As Reviewer #1 pointed out, however, we had to emphasise these goals clearer in the introduction, which we now did.

*I have only minor suggestions to improve clarity of figures (especially regarding choice of colors), and text. I recommend publication of the manuscript.*

*Fig. 2-. The profile colors are hard to distinguish, and likely would be near impossible for anyone with color-blindness. I suggest using more easily discernable colors.*

We have spent many thoughts on this topic and all colour scales in the manuscript have been chosen with the colour-blindness issue in mind. The colour scales we used (https://cran.r-project.org/web/packages/viridis/vignettes/intro-to-viridis.html) are perceptually uniform and robust to both colour-blindness *and* grey-scale printing. We would like to keep all those advantages. However, we do agree that the differences between colours are rather small in Fig. 2, which is partly because we left out yellow. You find the new version of the plot below, which includes yellow (thereby widens the colour gradient) and has bolder lines to make the colours more distinguishable. We hope that this improved version will find your approval.

[Figure]

*Fig. 2. Is oxygen available? I suspect it would be relevant especially for CH4.*

**Figure 2 already contains an oxygen profile. However, we do realize that we did not define "$cO_2$" because we do not need the term elsewhere and that it is rather easy to be confused with "$CO_2$". Therefore, we replaced the legend label with "$O_2$ concentration" to make it clearer (see above).**

*Line 41: 'two minima' are mentioned, but the subsequent text implies a minima during the spring bloom, and several subsequent minima throughout summer. Not clear when the second surface pCO2 minima typically occurs, or if there are several more? Perhaps revise 'two minima' to ': : :a minimum during spring and one or more subsequent minima throughout summer: : :'*

**Indeed, the two sentence parts do not really fit together and we revised the sentence according to your recommendation:**

**"The typical seasonality of surface carbon dioxide ($CO_2$) partial pressure ($p$$CO_2$) in the Baltic Sea features a minimum during spring and one or more subsequent minima throughout summer" (Line 37)**

*Lines-41-47 – could you clarify when the surface CO2 is typically under-saturated vs. super-saturated when describing the spring/summer surface pCO2?*

**We added the following to the end of the paragraph:**

**"and undersaturation is usually observed from March/April to September/October, depending on region (Schneider and Müller, 2018)." (Line 45).**

*Line 48-49 – '..vertical redox..' would be helpful to add oxygen profile to fig 2.*

**Please see above comment addressing Figure 2.**

*Fig. 6 –colors are difficult to distinguish.*

**As outlined above, the colour scales we used follow all criteria and many thoughts have been spent on them. We would like to keep it as is, since there are only three colours in this example. We made the legend labels bolder, though, as these were indeed a little hard to distinguish.**

*Line 332-335 You state that the estimated +0.4degK warming from atmospheric heat flux is insufficient to account for the observed warming. Can you remind the reader what the total surface warming was, and/or how much additional warming needs to be accounted for by mixing? (this could be included in this paragraph, or in the paragraph at the start of this section that describes temp relaxation after upwelling).*

**We clarified this and replaced "the observed warming of upwelled water masses" with "
[revised manuscript text omitted]

---

## Author Response (AR2)

We appreciate your comments and indicate where further clarifications were done. Line numbers given below refer to the manuscript with tracked changes.

5

The manuscript (ms) under review has been revised according to the comments of the two reviewers. Thus, the ms has been improved. However, there are still some points which should be addressed in a further revision.

**Major point**

The ms still reads like a technical report. The introduction has been rewritten substantially, however, the authors fail
to state the overarching scientific objectives(s) of their study. There is a list of points at the end of the Introduction, but this reads like a table of contents. The authors may want to rephrase this.

We can follow the line of thought of the reviewer. The list of points at the end of the introduction used to contain a mixture of both the contents and the objectives of our study. To ease the reader, we therefore split the contents and the aims (which we also expanded), which results in clearer and more concise statements. We keep the list

15 form to have both the approach and the aims of the study easily findable in the typeset text. More specifically, we replaced the old list with:

"In this study, we:

- present a method to identify upwelling events along the SOOP track based on wind and modelled SST data,
- compare upwelling-induced trace gas dynamics within several regions in the Baltic Sea,
  - examine the relaxation of upwelling events over time,
  - discuss interannual variability within the data set and highlight the importance of upwelling to understand CO2 and CH4 dynamics in the Baltic Sea,
  - evaluate whether a long-term trend can be inferred from the analysis of this eight-years data set taking into account the interannual variability, and
  - demonstrate the potential of extrapolating trace gas observations based on modelled SST data,

with the aim of:

- assessing the prevalence of upwelling and attributing observed trace gas signals to upwelling,
- revealing regional characteristics,
- explaining frequently observed features during and after upwelling events with a focus on underlying processes,
- identifying controlling mechanisms of seasonality and variability, and
- showing ways to estimate air-sea trace gas fluxes on a broader spatial scale under extremely variable conditions, which may be a useful method for other upwelling areas world-wide." (Lines 104–119)

**35**

20

25

30

Minor points (line number refer to the revised ms with 'track change' on.

Line (L) 8: What is meant by 'surface expression'. Please explain and rephrase.

We changed:

"Strong surface expressions of upwelling events..."

40 **to**:

"Large upwelling-induced SST decrease and trace gas concentration increase..." (Lines 7–8)

L20: 'addressing the environmentally important process of upwelling': At least for their study the authors explicitly state that they cannot estimate the importance of upwelling. Please rephrase.

**We cut "environmentally important". (Line 19) 45**

L32/33: 'leading to increased concentrations of reduced compounds': This is statement is wrong. In areas with ongoing upwelling we can measure enhanced concentrations of CO2 in the OMZ and thus in the upwelled waters found at the ocean surface. (you may check the figures in the ms.) CO2 cannot be classified as belonging to the group of 'reduced compounds'.

50

**We further specified by giving examples and changed:**

"This is a result of enhanced productivity and mineralisation, often hypoxic conditions in the subsurface waters leading to increased concentrations of reduced compounds, and an effective transport mechanism of these subsurface waters to the air-sea interface."

55 to:

> "This is a result of (1) enhanced productivity and mineralisation, (2) often hypoxic conditions in the subsurface waters leading to increased concentrations of reduced compounds such as CH4, N2O, NH4+, or H2S, and (3) an effective transport mechanism of these subsurface waters to the air-sea interface." (Lines 23-25)

Point (1) thereby indicates that both positive and negative  $pCO_2$  excursions are possible depending on the balance 60 of enhanced mineralisation and enhanced productivity.

L55/56: What is meant by 'offers the potential to study [...] early and with a high signal intensity'. Please explain and rephrase.

We elaborated this statement and added another reference. We changed:

[revised manuscript text omitted]

**130**

135

L152: replace 'near-atmospheric concentrations.' with 'near-atmospheric mole fractions.'.

**Done. (Lines 142-143)**

L483: '...other than variability using the methods we presented here.' What is meant by this? Please explain and rephrase.

**We changed:**

"longer trace gas time series are needed to detect trends other than variability using the methods we presented here."

to:

140 "longer trace gas time series are needed to not only detect variability, but also trends using the methods we presented here." (Lines 420–422)

L533: 'potentially greater impact on global trace gas fluxes than the Baltic Sea.' This statement needs a quantification of the trace gas emissions caused by upwelling in the Baltic Sea (and a comparison with data from other areas). However, the authors discuss why this is not possible (see L429-434 on page 19). So, please rephrase this statement.

**We changed:**

"which may also be used to study other upwelling areas with potentially greater impact on global trace gas fluxes than the Baltic Sea."

150 **to**:

145

"which may also be used to study other upwelling areas than the Baltic Sea." (Lines 471–472)

L550/551: 'other relevant effects'. This is a very vague statement. Please explain and give more details here.

We changed:

155 "while there appear to be other relevant effects especially towards the Gulf of Finland and around the island of Bornholm."

to:

160

"while there appear to be other relevant effects especially towards the Gulf of Finland (e.g. variability of the estuarine circulation) and around the island of Bornholm (e.g. lateral transport and CH4 release from the sediment)." (Lines 487–489)

4

**Upwelling-induced trace gas dynamics in the Baltic Sea inferred from 8 years of autonomous measurements on a ship of opportunity**

Erik Jacobs1, Henry C. Bittig1, Ulf Gräwe1, Carolyn A. Graves2, Michael Glockzin1, Jens D. Müller1, Bernd Schneider1, and Gregor Rehder1

[revised manuscript text omitted]